# CREW-Wildfire: Benchmarking Agentic Multi-Agent Collaborations at Scale

**Jonathan Hyun[1], Nicholas R Waytowich[2], Boyuan Chen[1]**
**[1]Duke University, [2]Army Research Laboratory**
http://www.generalroboticslab.com/CREW-Wildfire

**Reviewed on OpenReview:** https://openreview.net/pdf?id=8mr27qFzKR

## Abstract

Despite rapid progress in large language model (LLM)-based multi-agent systems, current benchmarks fall short in evaluating their scalability, robustness, and coordination capabilities in complex, dynamic, real-world tasks. Existing environments typically focus on small-scale, fully observable, or low-complexity domains, limiting their utility for developing and assessing next-generation multi-agent Agentic AI frameworks. We introduce CREW-WILDFIRE, an open-source benchmark designed to close this gap. Built atop the human-AI teaming CREW simulation platform, CREW-WILDFIRE offers procedurally generated wildfire response scenarios featuring large maps, heterogeneous agents, partial observability, stochastic dynamics, and long-horizon planning objectives. The environment supports both low-level control and high-level natural language interactions through modular PERCEPTION and EXECUTION modules. We implement and evaluate several state-of-the-art LLM-based multi-agent Agentic AI frameworks, uncovering significant performance gaps that highlight the unsolved challenges in large-scale coordination, communication, spatial reasoning, and long-horizon planning under uncertainty. By providing more realistic complexity, scalable architecture, and behavioral evaluation metrics, CREW-WILDFIRE establishes a critical foundation for advancing research in scalable multi-agent Agentic intelligence. All code, environments, data, and baselines will be released to support future research in this emerging domain.

## 1 Introduction

Coordinating multiple agents to solve complex, dynamic, and high-stakes tasks is a long-standing challenge in artificial intelligence (AI) (OroojlooyJadid & Hajinezhad, 2021; Ismail et al., 2018; Buşoniu et al., 2010). Real-world scenarios such as disaster response (Drew, 2021; Kashyap et al., 2025), autonomous infrastructure maintenance (Ismail et al., 2018; Krnjaic et al., 2024), workflow optimization (Krnjaic et al., 2024), or planetary exploration (Huang et al., 2020; Qin et al., 2025b;a) often require teams of agents with diverse capabilities to operate under uncertainty, partial observability, and real-time constraints. Among these, wildfire response (Seraj et al., 2021; Siedler, 2025) stands out as a prototypical example: a setting that demands scalable collaboration, heterogeneous roles, and adaptive decision-making over extended time horizons. As interest grows in generalist and open-ended AI systems (Ahn et al., 2022), there is a strong need to evaluate whether today's Agentic AI frameworks can meet the demands of such real-world complexity.

Traditionally, multi-agent systems have been studied through the lens of multi-agent reinforcement learning (MARL) (Buşoniu et al., 2010), decentralized control (Zhang et al., 2018), or classical robotics coordination (Perez et al., 2018). These approaches have yielded impressive capabilities in domains such as warehouse logistics (Krnjaic et al., 2024), robotic swarms (Huang et al., 2020), and cooperative manipulation (Chen et al., 2022). However, they typically rely on rigid communication protocols, domain-specific policies, or centralized planners (OroojlooyJadid & Hajinezhad, 2021) that limit generalization and flexibility. Moreover, these systems often struggle to scale in the number of agents or in the complexity of their operating environments

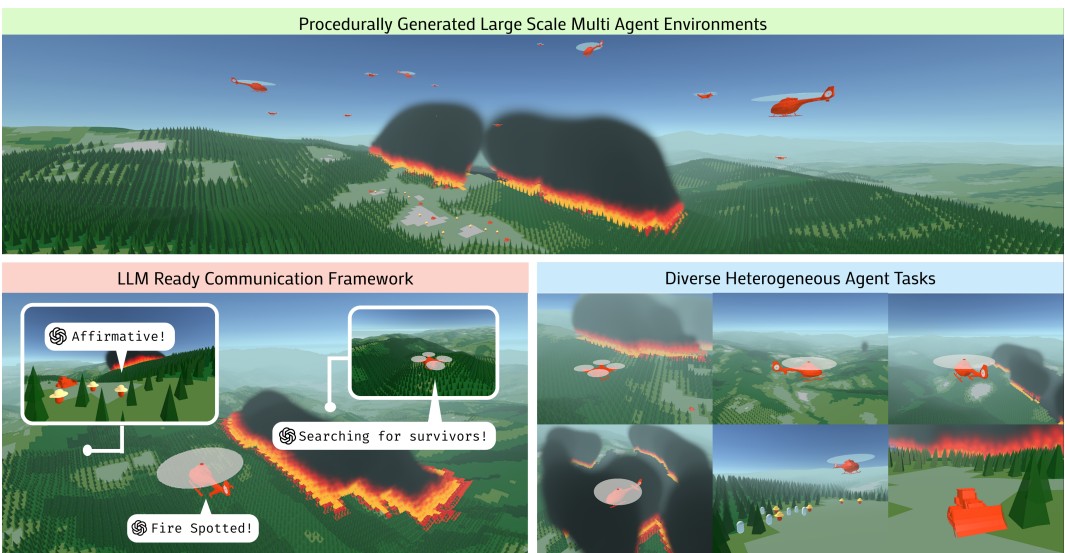

Figure 1: CREW-WILDFIRE features procedurally generated environments, an LLM-compatible multi-agent framework, and heterogeneous agents designed to evaluate Agentic collaborations at scale.

due to their inherent limitations in communication, commonly done through exchanging gradients (Buşoniu et al., 2010), observations (Foerster et al., 2016), or intentions (Kim et al., 2020). It becomes particularly challenging when coordination must emerge from high-level reasoning over extended periods of time rather than low-level behaviors.

Recent advances in large language models (LLMs) have opened a new frontier in Agentic AI (Wang et al., 2024; Zhang et al., 2024a; Park et al., 2023). LLM-based agents can communicate in natural languages, perform in-context reasoning, and dynamically coordinate using structured dialogue or shared goals. Early frameworks have demonstrated promising behaviors such as delegation, consensus-building, and role assignment, often in open-ended tasks like programming (Qian et al., 2024), board games (Chen et al., 2024a), or virtual social environments (Kaiya et al., 2023). However, these demonstrations largely operate in limited settings: they assume fully observable worlds, involve a small number of agents, and lack real-time embodiment or environmental complexity. As a result, it remains unclear whether existing multi-agent LLM frameworks can scale to realistic, physically grounded, long-horizon problems that demand both strategic coordination and low-level execution.

In this work, we introduce CREW-WILDFIRE (Fig. 1), an open-source benchmark specifically designed to evaluate Agentic multi-agent LLM systems under conditions of real-world scale and complexity. Built atop the human-AI teaming CREW Zhang et al. (2024c;d) simulation framework, CREW-WILDFIRE features procedurally generated wildfire environments with heterogeneous agents (e.g., drones, helicopters, bulldozers, and firefighters), partial observability, stochastic environments, and complex objectives such as civilian rescue, fire detection and suppression. It provides flexible observation and action spaces, ranging from low-level vectors to natural language commands, and includes built-in PERCEPTION and EXECUTION modules to interface with both low-level control primitives and LLM-based agents. To support rigorous evaluation, we propose a suite of task environments and behavioral goals, enabling both quantitative and qualitative assessment of multi-agent LLM frameworks on scalability, coordination, adaptability, and communication.

We implement and benchmark several recent LLM-based multi-agent Agentic frameworks in CREW-WILDFIRE. Our results reveal significant performance gaps and highlight open challenges. Our experiments show that while these systems exhibit emergent collaboration in simple tasks, they often fail to generalize to environments that require precise real-time coordination, spatial understanding, plan adaptation under uncertainty, and objective prioritization. By surfacing these limitations and offering a standardized evaluation platform, CREW-WILDFIRE provides a critical foundation for accelerating progress in large-scale Agentic intelligence.

Table 1: Feature comparison across different multi-agent embodied control environments.

| Environment | Maximum Agents | Heterogenous Agents | Low-Level Actions | Low-Level Observations | Partial Observability | Complex Long-Horizon Tasks | Stochastic Dynamics | Generative Environments |
|---|---|---|---|---|---|---|---|---|
| Lyfe Game (Kaiya et al., 2023) | 8 | ✗ | ✗ (Only discrete, high-level text) | ✗ (Only dialogues, text descriptions) | ✗ | ✔ | ✗ | ✗ |
| VirtualHome (Puig et al., 2018) | 7 | ✗ | ✔ (Discrete action tensors) | ✔ (Images) | ✗ | ✔ | ✗ | ✗ |
| PettingZoo (Terry et al., 2021) | 6 | ✗ | ✔ (Discrete + continuous action tensors) | ✔ (Low-level state vectors) | ✗ | ✗ | ✗ | ✔ |
| CUISINEWORLD (Gong et al., 2024) | 4 | ✗ | ✗ (Only discrete, high-level text) | ✗ (Only text descriptions) | ✗ | ✔ | ✗ | ✗ |
| PARTNR (Chang et al., 2024) | 2 | ✔ (2 agents) | ✔ (Continuous action tensors) | ✔ (Images) | ✔ | ✔ | ✗ | ✔ |
| **CREW-Wildfire (ours)** | **2000+** | ✔ (4 agents) | ✔ (Discrete + continuous action tensors) | ✔ (Images, ASCII, low-level states) | ✔ | ✔ | ✔ | ✔ |

In summary, our main contributions are:

- **A new open-source benchmark** for evaluating LLM-based multi-agent Agentic systems in procedurally generated, physically grounded, and high-stakes disaster response environments.

- **A suite of sub-environments and behavioral goals,** enabling fine-grained evaluation of collaboration, spatial reasoning, task delegation, and plan adaptation at scale.

- **Comprehensive benchmarking of state-of-the-art frameworks**, revealing performance gaps and providing open challenges for future research in scalable coordination and Agentic AI.

## 2    Related Works

**Benchmarks for Multi-Agent Embodied Environments.**    A wide range of simulation environments have been developed to benchmark multi-agent systems, spanning symbolic games, strategy simulations, and embodied tasks. Hanabi Bard et al. (2020) has served as a canonical benchmark for emergent communication and theory of mind under partial observability, but it is entirely symbolic and turn-based, lacking physical embodiment or dynamic interaction. StarCraft II Vinyals et al. (2017), in contrast, provides rich spatial dynamics and partial observability and has served as a challenging environment for micromanagement and tactical coordination. However, its use in multi-agent evaluation typically emphasizes unit-level control and scripted tasks, rather than open-ended planning in heterogeneous teams.

Environments such as Overcooked Carroll et al. (2019), PettingZoo Terry et al. (2021), Visual Hide and Seek Chen et al. (2020; 2021); Ji et al. (2025) introduce complex behaviors and embodiments but are limited in scale, environmental realism, and agent diversity. More recent efforts like MineDojo Fan et al. (2022) and VirtualHome Puig et al. (2018) explore open-ended tasks and embodiment, but primarily focus on single-agent settings or symbolic role-play scenarios.

Several simulation frameworks have also been adapted for real-world domains, including traffic control and disaster response. Notably, platforms like FireCommander Seraj et al. (2021) and Hivex Siedler (2025) simulate wildfire scenarios using multi-agent systems for resource management and disaster relief. While domain-relevant and complex, these environments often lack the architectural support for large-scale coordination and LLM-based agents. They do not support flexible agent communication protocols and high agent counts, making them unsuitable for evaluating the scalability of emerging Agentic AI frameworks.

CREW-WILDFIRE bridges these gaps by offering a fully open-source, embodied, and highly scalable environment centered on evaluating and benchmarking LLM-based multi-agent Agentic AI frameworks. It supports heterogeneous agents, complex and dynamic terrain, and realistic objectives under partial observability and stochastic conditions. By combining these features with support for low-level control and high-level language reasoning, CREW-WILDFIRE provides a unique testbed for evaluating large-scale collaboration, perception, and planning in Agentic multi-agent systems.

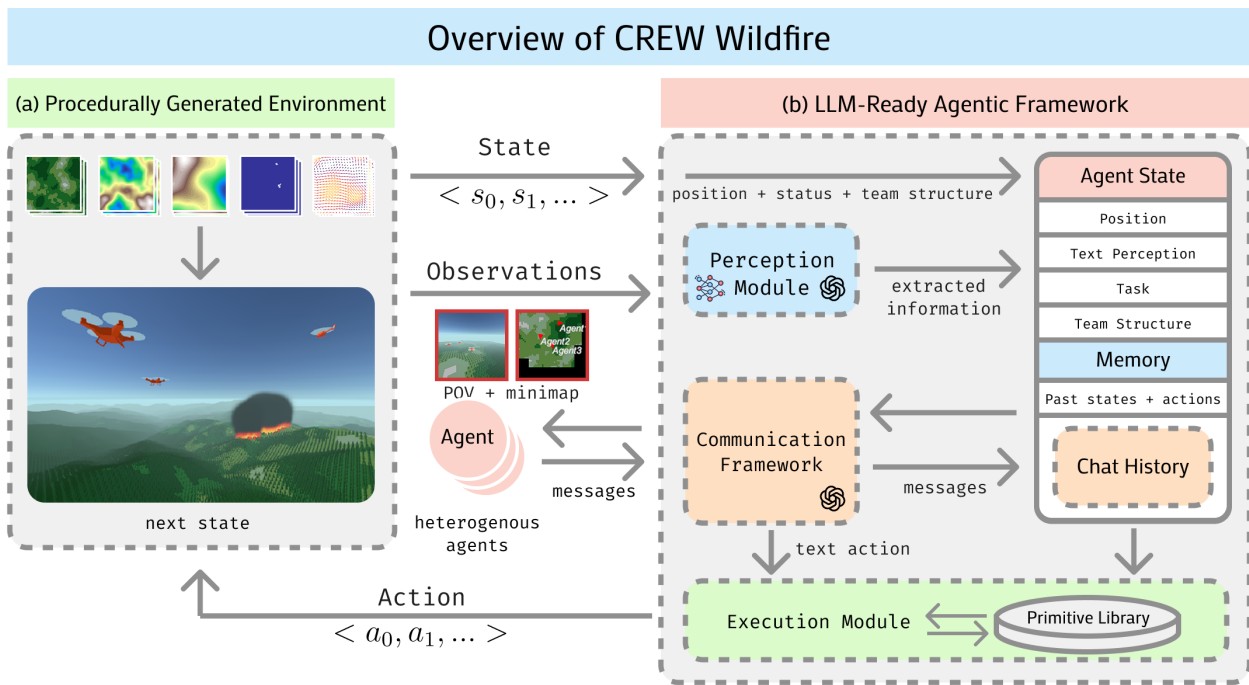

Figure 2: **An overview of the CREW Wildfire Framework**. (a) Environments are procedurally generated through a combination of Perlin noise textures modeling vegetation, elevation, moisture, settlement, and wind vector maps. (b) CREW-WILDFIRE supports template infrastructure generalized from existing LLM agent frameworks, including PERCEPTION and EXECUTION modules, built for easy implementation of multi-agent Agentic frameworks.

**Agentic AI and Multi-Agent Language-Based Frameworks.** The rise of LLMs has catalyzed a new wave of research in language-enabled Agentic AI. Methods such as CAMEL Li et al. (2023), ChatDev Qian et al. (2024), Lyfe Agents Kaiya et al. (2023), and Generative Agents Park et al. (2023) demonstrate that LLMs can coordinate in multi-agent settings through natural language communication, role assignment, and iterative planning. These systems have shown promising emergent behaviors such as delegation, consensus formation, and memory sharing. However, the vast majority of these demonstrations are under relatively constrained environments or purely symbolic domains, such as collaborative software development, dialogue-based games, single-agent tasks, or simplified simulations. They lack embodied complexity, team size scalability, continuous dynamics, and low-level control interfaces found in real-world systems.

CREW-WILDFIRE complements and extends these prior efforts by providing a scalable, grounded, and embodied benchmark specifically designed for LLM-based multi-agent tasks. It includes PERCEPTION and EXECUTION modules that enable language-based agents to interpret sensory information and generate executable actions, thereby supporting both high-level planning and low-level interaction. Furthermore, CREW-WILDFIRE introduces structured sub-environments and behavioral goals to facilitate rigorous evaluation of Agentic competencies such as task allocation, spatial reasoning, observation sharing, and adaptive planning—capabilities that are underexplored in current benchmarks.

## 3 The CREW-Wildfire Environment

### 3.1 Preliminaries

One natural path in designing a benchmark for large-scale LLM-based multi-agent systems is to adapt existing multi-agent reinforcement learning environments, such as StarCraft II or FireCommander, to support LLM-based agents. While this approach would enable direct usage of prior MARL benchmarks, it constrains the integration of flexible communication structures and high-level Agentic reasoning capabilities of language

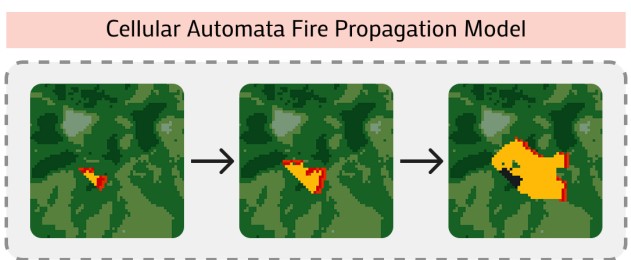

Figure 3: **The Cellular Automata Fire Propagation Model**

models. Another strategy is to extend current multi-agent LLM frameworks to handle larger, heterogeneous teams. However, these frameworks are typically built around simplified domains, and scaling agent count alone does not inherently introduce the strategic complexity or environmental diversity needed to challenge scalable Agentic intelligence.

Therefore, we chose to build CREW-WILDFIRE from the ground up. This approach allowed us to make principled design decisions tailored to evaluating Agentic intelligence at scale, including the development of heterogeneous agents, real-time coordination challenges, and long-horizon planning in dynamic environments.

**CREW Platform.** CREW-WILDFIRE is built on the CREW platform Zhang et al. (2024c), a scalable human-AI teaming simulation infrastructure using the Unity engine. Unity provides a visually rich and customizable development environment, while CREW provides flexible modules for rapid prototyping of complex game dynamics and AI-ready programming interfaces.

## 3.2 Environment Design

**Procedural Map Generation.** A distinctive feature of CREW-WILDFIRE is its scalable generative map capability. Each scenario is procedurally generated using Perlin noise to produce continuous terrain features (i.e., elevation, wind, moisture) and discrete land types (i.e., forest, brush, rock, water). Human settlements are also randomly placed controlled by seeds. These variables influence fire spread, visibility, and mission-critical decisions, ensuring each run presents unique and uncertain challenges. Visual illustrations of such procedure is shown in Appendix A.1 and Fig. 8.

**Wildfire Simulation.** We simulate fire dynamics using an advanced cellular automata model (Fig. 3) that incorporates slope, wind, vegetation type, moisture, and terrain features. This model creates realistic and unpredictable fire spread, forcing agents to adapt and exploit environmental features such as firebreaks. Specifically, we model the local fire spread probability $p_{\text{spread}}$ from a source cell to a neighboring target cell as:

$$p_{\text{spread}} = f(\text{slope}) \cdot \frac{\text{moisture}}{\text{moisture\_constant}} \cdot \left( \frac{\vec{w}}{\|\vec{w}\|} \cdot \frac{\Delta\vec{x}}{\|\Delta\vec{x}\|} + 1 \right)$$

where the slope factor $f(\text{slope})$ accounts for uphill versus downhill fire spread dynamics:

$$f(\text{slope}) = \begin{cases} \dfrac{e^{-k\cdot\text{slope}}}{2e^{-k\cdot\text{slope}} - 1}, & \text{if slope} < 0, \\ e^{k\cdot\text{slope}}, & \text{if slope} \geq 0. \end{cases}$$

Here, $\vec{w}$ denotes the wind vector and $\Delta\vec{x}$ is the direction vector pointing from the source cell to the target cell. The magnitude $\|\Delta\vec{x}\|$ represents the distance between cells: 1 for adjacent cells, $\sqrt{2}$ for diagonal cells. The slope factor $f(\text{slope})$ scales the influence based on terrain steepness, increasing spread probability uphill and decreasing it downhill. The moisture ratio is continuous, but ignition outcomes are binary (a cell either ignites or does not based on comparing $p_{\text{spread}}$ against a threshold). The moisture ratio attenuates the likelihood of ignition in wetter regions. The dot product modulates the alignment of wind and spread direction, increasing

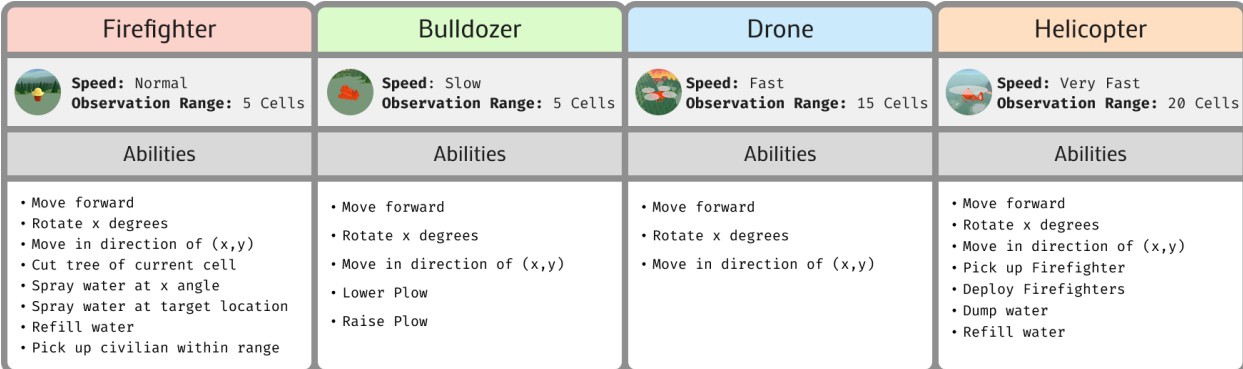

Figure 4: **The Four Heterogenous Agent Types**: Firefighters, Bulldozers, Drones, and Helicopters

fire spread probability when aligned and decreasing it when opposing. The propagation speed is determined by the game update frequency, which is user-configurable. This computed value $p_{\text{spread}}$ is then compared against a threshold to determine whether the target cell ignites in the next simulation step.

## 3.3 Agent Design

We include four heterogeneous agent types (Fig. 4) abstracted from real wildfire operations:

- **Firefighters** are generalist agents capable of cutting trees to create firelines, spraying water to extinguish fires or reduce flammability, and rescuing civilians.

- **Bulldozers** quickly clear flammable vegetation. However, they have limited speed and cannot directly extinguish or rescue civilians, requiring support from firefighter agents.

- **Drones** perform wide-area reconnaissance but lack physical intervention capabilities.

- **Helicopters** transport agents and civilians and deliver water. With large enough maps, helicopters play an indispensable role in transporting firefighters over distances too far to travel on foot.

These roles are designed to: 1) simplify but sufficiently represent real-world fire crew specializations, 2) span diverse skill sets that require cooperative execution, and 3) necessitate non-trivial coordination strategies due to inter-agent dependencies.

These agent types have complementary capabilities requiring coordination: 1) helicopters can transport firefighters over long distances but cannot rescue civilians independently, requiring helicopter-firefighter coordination, 2) drones can spot fires from afar while ground teams cannot, but drones cannot cut trees or extinguish fires, necessitating drone-ground coordination, 3) bulldozers can cut firebreaks efficiently but cannot rescue civilians, requiring coordination with firefighters.

CREW-WILDFIRE supports a large range of observation and action spaces, encompassing both low- and high-level control: Observations support direct mini-map encoding, third-person agent image data, textual descriptions of the observable environment, or ground-truth agent data vectors. Actions support a 3-dimensional discrete or continuous action tensor, textual descriptions of an action, or high-level primitives involving multi-step actions.

### 3.3.1 Perception and Execution Modules

Our PERCEPTION and EXECUTION modules are frequently found in and designed to be compatible with modern LLM-based multi-agent Agentic AI frameworks.

**Perception module** CREW-WILDFIRE provides various types of observations, such as ground-truth data tensors and raw images. However, LLMs still struggle with processing high-dimensional sensor data such

as images, and VLMs struggle with reasoning capabilities. Therefore, many multi-agent-LLM frameworks employ a PERCEPTION module that serves to bridge this gap. Our PERCEPTION module is made up of two components that first transform the raw tensor data into ASCII representations and then into high-level text summaries. The prompts for the PERCEPTION module can be found at Appendix A.2 along with some examples. We have observed that our PERCEPTION module can provide accurate and reliable observational information to the LLM agents, so that the agentic LLM framework can focus on team planning.

**Execution module** Consistent with related work (Zhang et al., 2024b), we found that while LLMs can succeed at high-level control, low-level control remains challenging for LLMs as of today. To address this limitation, many LLM-based agentic frameworks employ "execution modules" that serve as a middle layer between high-level planning and low-level control. Given that our core environment requires precise, discrete and continuous control, we developed our own built-in EXECUTION module to support these preexisting LLM-based frameworks while leaving additional flexibility for low-level control.

Our EXECUTION module uses LLMs to translate natural language commands into a series of one or more executable action codes. These action codes give all the necessary information to activate the associated low-level primitive functions (e.g., cutting a tree, spray water at a location, or pick up a civilian) within the simulation environment to control the agents. For information on these primitive functions, please refer to Appendix A.3. We also provide the prompts of the EXECUTION in Appendix A.4.

### 3.4   Pillars of CREW-Wildfire Design

Beyond the individual features described, CREW-Wildfire's strength lies in the combination of its properties, which together create a unique and challenging benchmark aimed at tackling the inherent challenges that a large and complex cooperative task, such as wildfire fighting, creates. We define the following pillars as essential to create a complex yet flexible environment that facilitates high-level cooperation: 1) **High scalability**: CREW-WILDFIRE can support a large number of agents (2000+ agents and 1 million+ cell maps on a desktop with 16GB GPU + 16GB RAM + 16GB GPU. Computational performance data up to 2000 agents and 1 million cells can be found in Appendix A.16.); 2) **Task complexity**: Due to the map randomness, fire simulation, and multiple map-dependent objectives, CREW-WILDFIRE does not always have a clear best solution that fits every case. As the task progresses, dynamics may change, which requires continual plan revision and adaptation; 3) **Heterogeneous agents**: Role heterogeneity demands coordination through complementary capabilities; 4) **Flexible observations and actions**: CREW-WILDFIRE accommodates a wide range of learning paradigms via modular observation and action layers.

These strengths make CREW-WILDFIRE a particularly challenging and comprehensive benchmark, where success would demonstrate a framework's ability to handle large-scale cooperation, partial observability, and long-term strategy all at once.

## 4   CREW-Wildfire Benchmarking Suite

Robust benchmarking in complex environments requires more than tracking cumulative rewards. It demands tools to diagnose where an algorithm is *conceptually challenged* (as revealed through behavioral competency analysis in Section 5.3), not just how well it scores. To address this, CREW-WILDFIRE Benchmarking Suite provides infrastructure for systematic evaluation, testing, and behavioral analysis.

**Procedurally Generated Task Levels.**   We define 12 distinct task levels (Table 2), four of which have both small and large size variants, resulting in 16 total benchmark configurations. Levels vary in team composition, map size, objectives, and difficulty. We measure the performance via task success, damage minimization, and agent/civilian safety. We also assign high-level behavioral goals to each level to evaluate a variety of high-level behaviors.

**Behavioral Goals.**   We designed 7 behavioral competencies, ranging from simple task designation to complex objective prioritization. Since each level is marked with behavior goals, we can use them to pinpoint

Table 2: **Twelve Benchmark Levels (with size variants for four levels)**. F: Firefighters, B: Bulldozers, D: Drones, H: Helicopters

| Name | Objective | Agents | Map Size | Max Score | Behaviors |
|---|---|---|---|---|---|
| **Cut Trees** Scoring Function: Trees cut in labeled cells/lines | | | | | |
| Sparse (small) | Cut all trees in labeled cells | 3 F | 30 | 18 | TD |
| Sparse (large) | Cut all trees in labeled cells | 10 F | 60 | 75 | TD |
| Lines (small) | Cut all the labeled lines of trees | 2 F, 1 B | 30 | 30 | TD, AC |
| Lines (large) | Cut all the labeled lines of trees | 4 F, 3 B | 60 | 105 | TD, AC |
| **Scout Fire** Scoring Function: Drones over the fire, max of two | | | | | |
| (small) | Scout and confirm a fire within the map | 3 D | 100 | 2 | TD, SR, OS |
| (large) | Scout and confirm a fire within the map | 5 D | 250 | 2 | TD, SR, OS |
| **Transport Firefighters** Scoring Function: Firefighters at target locations | | | | | |
| (small) | Transport all firefighters to a target location | 6 F, 1 H | 100 | 6 | AC, SR, RC |
| (large) | Transport all firefighters to a target location | 12 F, 2 H | 250 | 12 | AC, SR, RC |
| **Rescue Civilians** Scoring Function: Civilians at target location | | | | | |
| Known Location (small) | Rescue all civilians to a target location | 3 F | 40 | 3 | TD, SR, PA |
| Known Location (large) | Rescue all civilians to a target location | 3 F | 40 | 9 | TD, SR, PA |
| Search and Rescue | Locate and rescue all civilians to a target location | 5 F, 2 D | 100 | 5 | TD, SR, OS, PA |
| Search + Rescue + Transport | Locate and rescue all civilians to a target location | 10 F, 2 D, 2 H | 150 | 10 | TD, AC, SR, OS, RC, PA |
| **Suppress Fire** Scoring Function: Trees Destroyed + 20×Agents Lost | | | | | |
| Extinguish | Extinguish the fire at a known location with water | 8 F | 60 | N/A | TD, SR, PA |
| Contain | Contain the fire at a known location without water | 5 F, 1 B | 60 | N/A | TD, AC, SR, PA |
| Locate and Suppress | Suppress the fire at an unknown location | 5 F, 1 B, 2 D | 100 | N/A | TD, AC, OS, SR, PA |
| Locate + Deploy + Suppress | Suppress the fire at an unknown location | 10 F, 2 D, 2 H | 150 | N/A | TD, AC, OS, SR, RC, PA |
| **Full Environment** Scoring Function: Trees Destroyed + 20×Agents Lost + 100×Civilians Lost | | | | | |
| Full Environment | Locate and Suppress the fire while rescuing civilians | 10 F, 1 B, 2 D, 2 H | 200 | N/A | All |

emerging collaborative behaviors. For instance, the `Cut Trees` levels require `Task Designation` for agents to split up the trees to cut.

The behavioral goals include: 1) `Task Designation` (**TD**): the ability to divide tasks among agents; 2) `Agent Capitalization` (**AC**): the ability to recognize and capitalize on the strengths and weaknesses in heterogeneous teams; 3) `Spatial Reasoning` (**SR**): The ability to reason and plan accordingly with spatial information; 4) `Observation Sharing` (**OS**): the ability to communicate useful observations when necessary; 5) `Realtime Coordination` (**RC**): the ability to communicate and rely on other agents to perform synchronized tasks; 6) `Plan Adaptation` (**PA**): the ability to adapt and revise plans; 7) `Objective Prioritization` (**OP**): the ability to rank competing goals contextually. More information is available in Appendix.8

# 5 Benchmark Experiments

## 5.1 Experiment Setup

Our goal is to assess how well current LLM-based multi-agent Agentic frameworks scale and generalize in realistic, embodied, and partially observable environments. We selected four representative frameworks from recent literature that span approaches ranging from fully decentralized to hybrid centralized-decentralized systems:

**CAMON** Wu et al. (2024) employs a hybrid coordination scheme with a dynamic leader who issues task assignments and global updates. Leadership transfers occur through agent-initiated interactions.

**COELA** Zhang et al. (2024b) is a decentralized structure where each agent independently proposes information, evaluates the necessity of communication, and performs actions based on its generated plans.

**Embodied** Guo et al. (2024) follows a decentralized structure with no leader agent. Agents alternate communication rounds, allowing message broadcasting or targeted exchanges, followed by independent action planning.

**HMAS-2** Chen et al. (2024b) adopts a hybrid coordination scheme, using a centralized planner to propose actions refined through distributed agent feedback until consensus is reached.

Table 3: **Scores across All Baseline on Twelve Levels** (mean ± standard deviation).

| | Max Score | CAMON | COELA | Embodied | HMAS-2 | Do Nothing |
|---|---|---|---|---|---|---|
| Cut Trees: Sparse (Small) | 18 | **18.00±0.00** | 14.00±4.18 | 14.60±2.07 | 17.40±0.89 | 0.00±0.00 |
| Cut Trees: Sparse (Large) | 75 | **72.00±4.00** | 57.67±25.00 | 50.33±9.00 | 56.33±13.00 | 0.00±0.00 |
| Cut Trees: Lines (Small) | 30 | **30.00±0.00** | **30.00±0.00** | **30.00±0.00** | 28.00±2.65 | 0.00±0.00 |
| Cut Trees: Lines (Large) | 105 | **94.33±7.00** | 83.67±24.00 | 90.33±1.00 | 90.33±6.00 | 0.00±0.00 |
| Scout Fire (Small) | 2 | **1.60±0.89** | 0.00±0.00 | 0.80±0.84 | 1.20±1.10 | 0.00±0.00 |
| Scout Fire (Large) | 2 | 0.00±0.00 | 0.00±0.00 | 0.00±0.00 | 0.00±0.00 | 0.00±0.00 |
| Transport Firefighters (Small) | 6 | **6.00±0.00** | 3.00±5.00 | 4.60±4.00 | 4.40±5.00 | 0.00±0.00 |
| Transport Firefighters (Large) | 12 | **10.00±0.00** | 8.33±5.00 | 9.33±2.00 | 8.33±5.00 | 0.00±0.00 |
| Rescue Civilians: Known Location (Small) | 3 | **3.00±0.00** | 0.67±0.58 | 2.67±0.58 | 2.33±0.58 | 0.00±0.00 |
| Rescue Civilians: Known Location (Large) | 9 | 4.00±5.00 | 0.00±0.00 | **4.33±3.00** | 4.00±2.00 | 0.00±0.00 |
| Rescue Civilians: Search and Rescue | 5 | 0.00±0.00 | 0.00±0.00 | 0.00±0.00 | 0.00±0.00 | 0.00±0.00 |
| Rescue Civilians: Search + Rescue + Transport | 10 | 0.00±0.00 | 0.00±0.00 | 0.00±0.00 | 0.00±0.00 | 0.00±0.00 |
| Suppress Fire: Extinguish | 0 | −593.33±196.51 | −635.00±189.65 | −636.33±193.67 | −624.33±196.92 | **−519.67±174.91** |
| Suppress Fire: Contain | 0 | −736.67±200.04 | −751.33±202.60 | −757.33±177.42 | −739.00±199.96 | **−660.67±184.50** |
| Suppress Fire: Locate and Suppress | 0 | −1062.67±248.61 | −1062.67±248.61 | −1062.67±248.61 | −1062.67±248.61 | −1062.67±248.61 |
| Suppress Fire: Locate + Transport + Suppress | 0 | −729.67±387.07 | −729.67±387.07 | −729.67±387.07 | −729.67±387.07 | −729.67±387.07 |
| Full Environment | 0 | −5571.67±3347.06 | −5516.00±3348.81 | −5539.33±3381.69 | −5522.67±3358.20 | **−5502.67±3330.02** |

**Do Nothing** is introduced as a naive baseline where no agent intervenes in the environment.

All baselines are fully zero-shot, LLM-based multi-agent frameworks with no learning during inference. These implementations follow their original published designs without training or fine-tuning.

**Evaluation Protocol.** To ensure consistency across these frameworks, we implemented compatible PERCEPTION and EXECUTION modules to enable uniform testing.

We use GPT-4o as the underlying model, with temperature set to 0 for deterministic results and a single completion per decision step. Experiments were conducted on a laptop with a 3.0 GHz CPU, RTX 3060 GPU, and 16GB RAM. We ran between 3-10 random seeds on all 16 level configurations for a total of 410 trajectories. Seeds and other hyper-parameters are located in A.19. We measured task success, task duration, API call frequency, and input/output token usage. Results are in Tab.3.

We used GPT-4o due to its superior performance on spatial reasoning and task planning among popular benchmarks at the time of experiments. Therefore, we chose to use it to benchmark all baselines with the same LLM backbone. Moreover, our focus is not on language model capabilities in the baselines, but instead, on different multi-agent LLM planning architectures and workflows and their performance on complex and large-scale environments. Finally, GPT-4o was also used by the original implementations in the baseline studies, hence our choice here for the benchmark. We leave the exploration of different LLM models as future work.

## 5.2 Results and Findings

**Current frameworks handle simple tasks but falter at scale.** Most frameworks, especially CAMON, performed well on `Cut Trees` tasks, showcasing effective task designation and asynchronous execution. In smaller maps, most frameworks successfully divided the work, as seen in (Fig.5). However, in larger instances, decentralized systems such as COELA and Embodied performed worse due to task overlap, with multiple agents redundantly cutting the same tree due to the lack of shared global knowledge.

**Coordination breaks down on complex and large-scale tasks.** In tasks such as `Suppress Fire` and `Search and Rescue`, performance across all frameworks was poor, often worse than the Do Nothing baseline due to penalties from agent loss. Centralized and hybrid systems like CAMON and HMAS-2 with leader agents struggled to assign roles effectively in complex scenarios, often issuing overlapping or redundant plans. While leader agents could successfully coordinate low-level tasks, such as cutting lines of trees (Tab. 3), they failed to perform the same level of task decomposition in complex scenarios. In Fig. 6(a), multiple agents were directed to cut the same tree without further decomposing the broader firebreak construction plan, showing that leader agents struggle to make complex multi-level tasks.

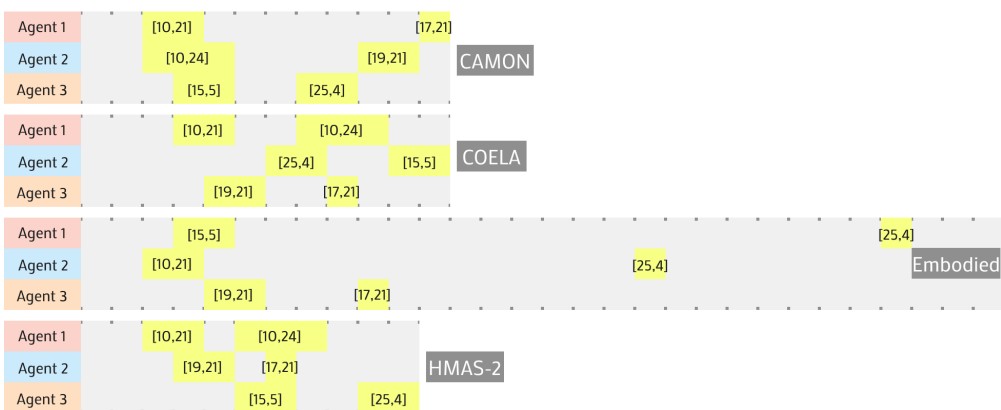

Figure 5: **Sample Multi-Agent Trajectories** for the `Cut Trees: Sparse(small)` level. Each highlighted section represents the correct tree being cut by that agent. Horizontal direction denotes time progression from left to right.

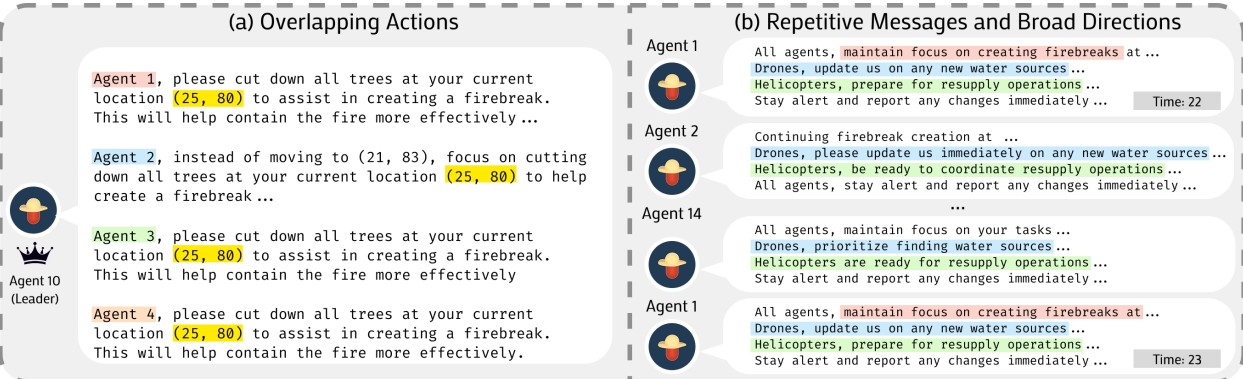

Figure 6: **Common Failure Cases.** (a) Overlapping actions in CAMON due to failed subtask decomposition by the leader during firebreak creation. (b) Repetitive and vague messaging in Embodied.

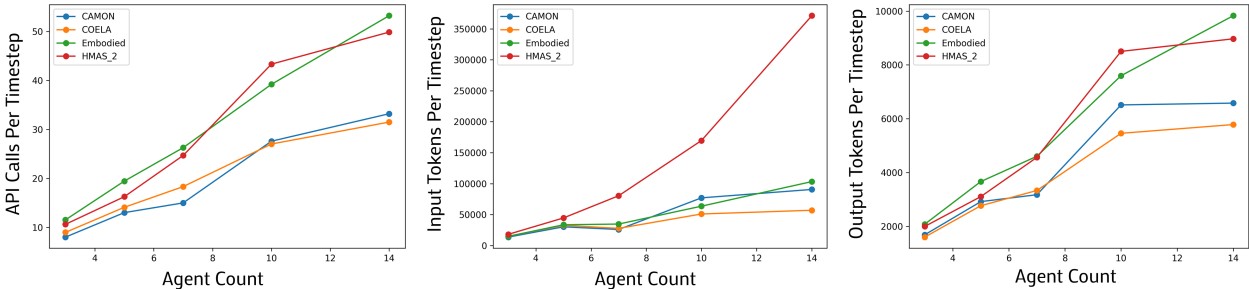

Figure 7: API calls, input tokens, and output tokens per timestep across agent counts. These metrics show algorithm-side operations only; the CREW-Wildfire environment itself has zero API cost as it is entirely self-contained.

Meanwhile, decentralized frameworks such as COELA and Embodied faced noise and repetition. As seen in Fig. 6(b), agents issued vague, overlapping instructions and failed to converge on a detailed strategy. Agents all wanted to take charge of the entire plan ending up in a rigid loop without adaptation.

Table 4: **Behavior Competency Scores (BCS) per algorithm.** Objective Prioritization is omitted from our competency analysis because no baseline encountered both the active fire front and endangered civilians within the full-environment level across our recorded trajectories.

| Behavior | CAMON | COELA | Embodied | HMAS-2 |
|---|---|---|---|---|
| Task Designation (TD) | **0.454** | 0.279 | 0.381 | 0.399 |
| Agent Capitalization (AC) | **0.474** | 0.391 | 0.435 | 0.418 |
| Spatial Reasoning (SR) | **0.368** | 0.156 | 0.301 | 0.301 |
| Observation Sharing (OS) | **0.180** | 0.067 | 0.124 | 0.153 |
| Realtime Coordination (RC) | **0.430** | 0.305 | 0.374 | 0.352 |
| Plan Adaptation (PA) | **0.239** | 0.093 | 0.219 | 0.209 |

**Communication costs scale with agent count.** As shown in Fig. 7, token usage scales with agent population. API calls and output tokens increase roughly linearly. However, input token usage in HMAS-2 grows quadratically, due to its shared observation vector that scales with the number of agents and partial observability. In a fully observable environment with a small size, it is feasible to provide each agent all the environment information. However, in large environments like ours ($1000 \times 1000$ cells), this makes universal context sharing inefficient. These results highlight the need for scalable architectures, e.g., decentralized or hierarchical approaches, that avoid token blow-up through more structured information routing.

### 5.3 Behavioral Analysis

To better evaluate the competency of current multi-agent agentic systems beyond task completion, we introduce the **Behavior-Competency Score (BCS)**, which captures the average normalized success across all tasks associated with a given high-level behavioral goal. With BCS, we can extract more interpretable and quantifiable insights into how the structures of different architectures facilitate different behaviors and where, on a behavior level, they succeed or fail.

**Step 1: Level Normalization.** The range of raw scores across different task levels is different due to different scoring functions. Therefore, we must first normalize our scores across all tasks types.

**Finite (reward) tasks** are tasks such as `Cut Trees`, where the team receives positive scores, such as for every correct tree cut. **Open-ended (penalty) tasks** are tasks such as `Suppress Fire`, where the team receives open-ended negative scores, such as for every tree burned.

To normalize task $\ell$, we first find the baseline ($B_\ell$) or the worst possible score on a task (0 for **finite** tasks and the Do-Nothing score + the penalty for losing all agents for **open-ended** tasks). Then we find the target ($T_\ell$) or the best possible score on a task (max reward for **finite** tasks and 0 for **open-ended** tasks). Then, we normalize the algorithm's raw score ($s_{a,\ell}$) by subtracting the baseline and dividing by the range. Lastly, we logarithmically scale scores for open-ended tasks to amplify the smaller improvements over the baseline score.

$$\text{NS}_{a,\ell} = \begin{cases} \dfrac{s_{a,\ell} - B_\ell}{T_\ell - B_\ell}, & \text{for finite (reward) tasks;} \\[2em] \dfrac{\log\left(1 + \frac{s_{a,\ell} - B_\ell}{T_\ell - B_\ell}\right)}{\log 2}, & \text{for open-ended (penalty) tasks.} \end{cases}$$

**Step 2: Behavioral Aggregation.** Since each benchmark task is associated with a set of behavioral goals, each behavioral goal $g$ also has a set of tasks $\mathcal{T}_g$ that possess it as a behavior. Therefore, the $\text{BCS}_{a,g}$ for a given behavioral goal $g$ and algorithm $a$ is the mean of the normalized scores of that set.

$$\text{BCS}_{a,g} = \frac{1}{|\mathcal{T}_g|} \sum_{\ell \in \mathcal{T}_g} \text{NS}_{a,\ell}$$

**BCS Results and Findings**  Overall, as shown in Tab. 4 and Fig. 10, current methods perform well at Realtime Coordination (RC), Task Designation (TD), and Agent Capitalization (AC), with CAMON particularly standing out across all behaviors. However, while all methods score comparably in behaviors such as Realtime Coordination (RC) and Agent Capitalization (AC), their performance varies much more in others, including Observation Sharing (OS), where the decentralized COELA scores significantly lower than the rest. Perhaps this implies that although dynamic communication may help planning and real-time coordination, periodic communication phases, such as those in Embodied and HMAS-2, are better suited for regular observation sharing.

The Plan Adaptation (PA) BCS also remains low across all baselines. PA is tested most extensively in `Suppress Fire` tasks, where fire shifts unpredictably. In these levels, instead of adapting when circumstances change, teams often reaffirm early decisions and plans, fixing agents to an outdated course of action. For instance, if fire lines breached an established boundary, most agents persisted in their outdated trajectories, which often resulted in them being destroyed by the new danger.

### 5.4  Outlook

Our study reveals that while existing multi-agent LLM frameworks demonstrate promising behaviors in simple, structured tasks, they struggle to generalize to complex, large-scale environments that demand dynamic coordination, role specialization, hierarchy, and real-time adaptation. These findings point to several critical directions for future research:

**Scalable Architectures and Efficient Algorithms:** More scalable communication architectures are needed to manage information flow as agent populations grow, such as hierarchical or attention-based routing. Our benchmark reveals efficiency challenges in current algorithms when scaled to complex tasks, inspiring algorithm development that prioritizes both scalability and computational efficiency.

**Adaptive Planning and Reasoning:** Effective abstraction and modularity in task planning may enable agents to better decompose and assign sub-goals in evolving scenarios. Future systems should integrate uncertainty-aware reasoning and adaptive leadership to respond flexibly to noisy or incomplete observations.

**Evaluation and Human-AI Teaming:** Developing systematic methods to measure communication efficiency, redundancy, and coordination-specific metrics for large-scale multi-agent problems remains an open challenge. We will open-source all recorded observations, actions, and agent communications to enable deeper trajectory analysis. Additionally, while our environment supports low-level human control, studying human performance requires investigating complex human teaming dynamics with 10+ agents in large-scale dynamic tasks, which we plan to explore in future work.

## 6  Limitations

We note several aspects of CREW-WILDFIRE that can be further improved. First, despite heavy referencing to real-world heterogeneous setups of multi-agent collaborations and coordination applications, our abstraction of agent roles is not yet a one-to-one mapping of real-world wildfire complexity and agent ability, potentially limiting direct generalizability and deployment in real-world wildfire responses. The low-level control is also abstracted, excluding the real-world physics and actuator simulation in robotics simulators. We intentionally simplify these aspects for our initial developments to focus on the planning components. This design choice also allows controlled and repeatable benchmark efforts targeting current LLM-based Agentic frameworks. CREW-WILDFIRE still remains the state-of-the-art scale and complexity in LLM-based multi-agent Agentic benchmarks. We plan to incorporate real-world embodied agents into CREW-WILDFIRE in our future iterations.

Second, despite the large number of experiments across multiple random seeds, baselines, and task levels, more runs on different prompt variations can provide further insights on the subtle but important system designs. Our current experiments are limited by the relatively high costs associated with LLM token usage. For example, in algorithms where agents are given a global-state vector (the combination of each agent's perceptions), the input token cost for that step scales on the order of $N^2$. With the cost of LLM-query

becoming more affordable, CREW-WILDFIRE will provide a promising evaluation platform to study more detailed algorithm designs.

Moreover, the above limitations are not a result of the CREW-WILDFIRE as a benchmark, but rather the current algorithmic token inefficiencies that limit the extent of realistic scalability testing. The lack of baseline data with the full capabilities of CREW-Wildfire [$2,000+$ agents and $1,000,000+$ cell map size] is due to the infeasibility resulting from the existing algorithm constraints, as indicated by our extensive quantitative and qualitative experiments. We include computational performance data up to 2000 agents and 1 million cells in Appendix A.16.

## 7 Conclusion

We introduced CREW-WILDFIRE, an open-source benchmark designed to evaluate the scalability and robustness of multi-agent systems powered by LLMs. Through procedurally generated environments including heterogeneous agents, partial observability, and long-horizon coordination, the benchmark exposes the limitations of current frameworks in handling complex, real-time, and high-agent-count tasks. Our experiments reveal critical performance gaps in role assignment, communication efficiency, and adaptive planning. While there are limitations in our scope and baseline data, by releasing the environment, sub-tasks, and baseline implementations, we aim to provide a foundation for the community to develop and compare next-generation Agentic systems. While the deployment of autonomous LLM agents presents positive opportunities for decision-making efficiency, it also carries significant ethical risks, necessitating robust frameworks and monitoring via open-source oversight to prevent misuse and ensure accountability.

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

# Appendix

## A.1 Map Generation and Evaluation Process

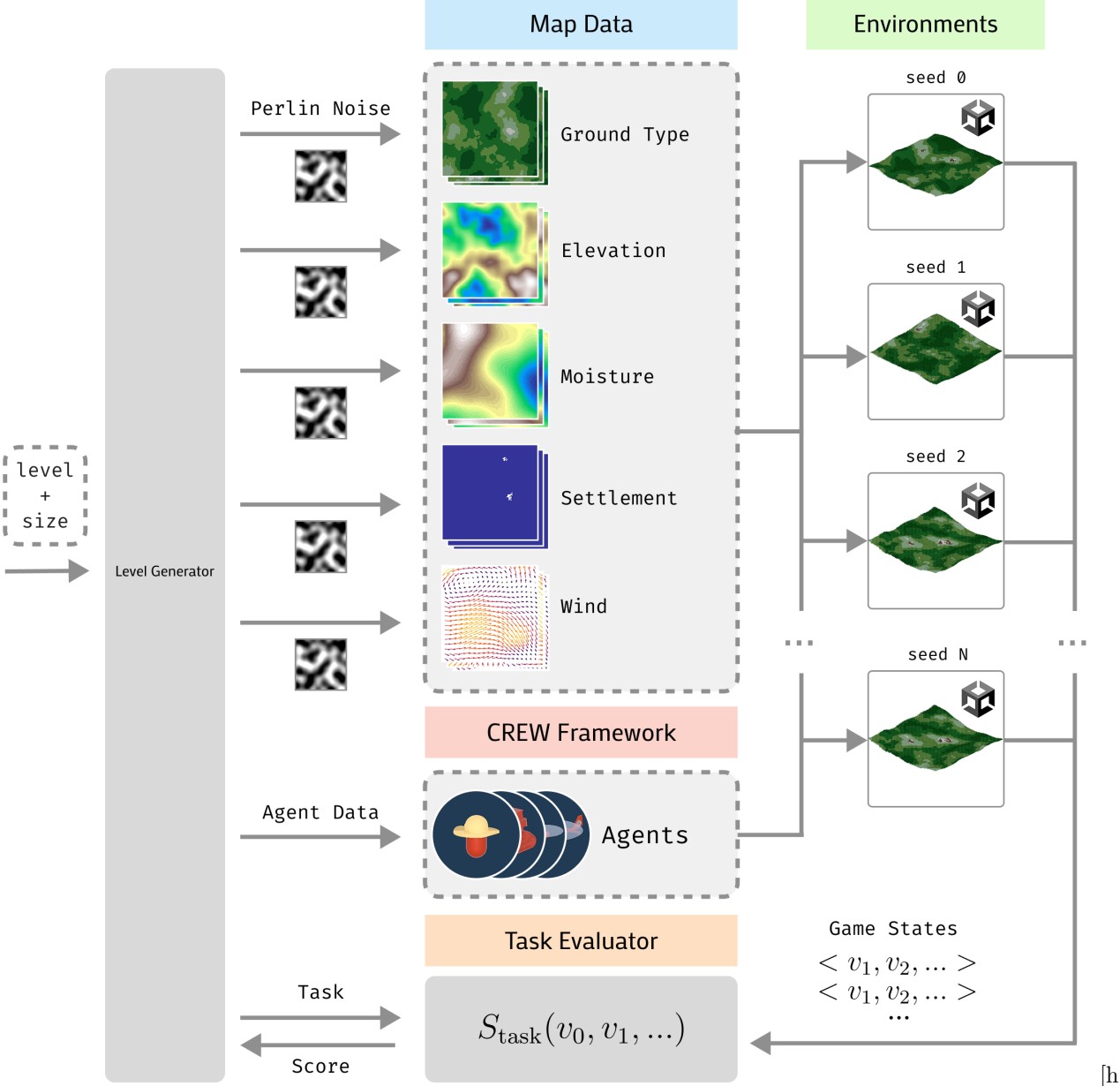

Figure 8: Map generation and Evaluation process. Given the chosen level and level size, the Level Generator creates 5 distinct Perlin Noise textures for each generation seed: one for Ground Type, Elevation, Moisture, Settlement, and Wind X + Y vectors. After scaling or clamping them into appropriate maps, they are put together as the generated environments. The Task Evaluator then repeatedly checks their state vectors to return a score depending on task type.

## A.2 Perception Module

The following prompt summarizes the ASCII representation.

---

**Prompt for PERCEPTION Module**

You are AGENT X, and your current location is POSITION, and thus your minimap view will be the range $X : [x_0 - x_1]$, $Y : [y_0 - y_1]$, where $x_0 = x - \text{range}/2$ and $x_1 = x + \text{range}/2$ (and similarly for y coordinates), centering the minimap on the agent's location, with the top corner of the map being (0,0).

This is your minimap view:

[ASCII ENCODED MINIMAP]

Each cell is represented by a character corresponding to the type of terrain:
0: brush (no trees)
1: light forest (1 tree)
2: medium forest (2 trees)
3: dense forest (3 trees)
i: Ignited
f: On Fire
e: Extinguishing
x: Fully Extinguished
w: Water Source Cell (no trees)
B: building (no trees)

IGNORE ALL "-". Those are unrevealed cells. They will reveal themselves when you get closer to them.

The cells in single quotations are wet cells. 'C' cells are civilians.

The bolded cell is the current cell you are in. It is a X cell at POSITION. There are other nearby agents at:

[OTHER AGENT LOCATIONS]

Your job is to process and understand your surroundings. Do not directly report explicit information from the minimap, but rather spatially understand your surroundings. Do not refer to character representations of the minimap, only what they actually represent. Report general observations in general directions. Also report if there are specific cells of interest, such as fires, civilians, water, etc. If there are any, calculate their exact locations by explicitly counting cells. You should return a detailed but concise text summary paragraph of all relevant information, including location, surroundings, and presence of important cells.

---

1. For example, the following raw observations input:

```
[{'location': (33,93), "trees": 2, "civilians": false, "status": burnable},
 {'location': (34,93), "trees": 3, "civilians": false, "status": burnable},
 {'location': (35,93), "trees": 3, "civilians": false, "status": burnable},
 ... (397 more items)]
```

2. Is transformed into:

```
2,3,3,3,3,3,3,3,3,2,2,2,2,2,2,3,3,2,2,2,3,
2,3,3,3,3,2,2,2,2,2,2,2,2,2,2,2,2,2,2,2,3,
2,2,3,3,3,2,2,2,2,2,2,2,2,2,2,2,2,2,2,2,3,
2,2,3,3,2,2,2,2,2,2,2,2,2,2,2,2,2,3,3,2,3,
2,2,2,3,2,2,2,2,2,2,1,1,1,2,2,2,2,3,3,2,2,
2,2,2,2,2,2,2,2,2,1,1,1,1,1,2,2,2,2,2,2,2,
2,2,2,2,2,2,1,1,1,1,1,1,1,1,2,2,2,2,2,2,2,
2,2,2,2,2,2,1,1,1,1,1,1,1,1,2,2,2,2,2,2,2,
2,2,2,2,2,1,1,1,1,1,1,1,1,1,1,1,1,1,1,1,2,
2,2,2,2,1,1,1,1,1,1,1,2,1,1,1,1,1,1,1,1,1,
2,2,2,2,2,1,1,1,1,1,1,*1*,1,1,1,1,1,1,1,1,1,1,
2,2,2,2,2,1,1,1,1,1,1,2,1,1,1,1,1,1,1,1,1,
1,2,2,1,1,1,1,1,1,1,1,2,1,1,1,1,1,1,1,1,1,
1,1,1,1,1,1,1,1,1,1,1,1,1,1,1,1,1,1,1,1,1,
1,1,1,1,1,1,1,1,1,0,1,1,1,0,0,0,0,0,0,0,0,
1,1,1,1,1,1,0,0,0,0,0,0,0,0,0,0,0,0,0,0,0,
0,1,1,1,1,1,0,0,0,0,0,0,C,0,0,0,0,0,0,0,0,
0,1,1,1,1,1,1,1,0,0,0,C,0,C,0,0,0,0,0,0,0,
0,1,1,1,1,1,1,0,0,0,0,0,0,0,0,C,0,0,0,0,0,
1,1,1,1,1,1,1,1,1,0,0,0,0,0,0,0,0,0,0,0,0,
1,1,1,1,1,1,1,1,1,0,0,0,0,0,0,0,0,0,0,0,0,
```

3. Which is summarized into:

I am currently positioned at (43, 103) within a light forest area. My immediate surroundings are predominantly light forest, with a mix of medium and dense forest areas to the north and northwest. To the north and northwest, there are patches of medium and dense forest, indicating a potentially challenging terrain for movement and visibility. The southern and southeastern areas transition into brush and open spaces. Significantly, there are civilians present in the southern region of my view. One civilian is located at (43, 110), and three more are at (45, 110), (46, 109), and (48, 111). These civilians are in an area transitioning from light forest to brush. There are no immediate signs of fire or ignited areas within my visible range.

### A.3 Primitives

Primitive functions serve as a simplified way to control our agents' low-level behavior within the environment. Although it is possible to control each agent's movement or rotation manually at each step, these primitives allow agents to focus on high-level planning instead of struggling with tedious low-level control. This is especially useful, since LLMs are proven to struggle with low-level control.

For example, the primitive 'move to location' may involve a multi-step sequence of movement and rotation actions to move the agent to the desired location.

Here is the list of all primitive methods agents can use within the environment:

Table 5: Primitives + Actions for All Agent Types

| Primitive Name | Parameters | Description |
|---|---|---|
| **Firefighter Primitives** | | |
| Move to Location | <**1**, x coord, y coord> | Continually move in the direction of the location until it reaches there |
| Cut $X$ Trees in current cell | <**2**, # of trees, 0> | Cut a single tree $X$ times |
| Cut All Trees in current cell | <**3**, 0, 0> | Cut a single tree until there are 0 trees left |
| Pick Up Civilian | <**4**, 0, 0> | Pick up closest civilian |
| Drop Off Civilian | <**5**, 0, 0> | Unload carried civilian to current cell |
| Spray Water Cone toward target | <**6**, x coord, y coord> | Aim at target, spray water in a wide cone |
| Refill Water | <**7**, 0, 0> | Refill water supply if over a water source |
| **Bulldozer Primitives** | | |
| Drive to Location (No Cut) | <**1**, x coord, y coord> | Continually move in the direction of the location without the plow lowered until it reaches there |
| Drive to Location (Clear Path) | <**2**, x coord, y coord> | Continually move in the direction of the location with the plow lowered until it reaches there |
| **Drone Primitives** | | |
| Fly to Location | <**1**, x coord, y coord> | Continually fly in the direction of the location until it reaches there |
| **Helicopter Primitives** | | |
| Fly to Location | <**1**, x coord, y coord> | Continually fly in the direction of the location until it reaches there |
| Pick Up Firefighters | <**2**, 0, 0> | Load nearby firefighter agents |
| Drop Off Firefighters | <**3**, 0, 0> | Unload all carried firefighters to current cell |
| Refill Water | <**4**, 0, 0> | Replenish water tank if over a water source |
| Drop Water | <**5**, 0, 0> | Deploy one water payload at current location |

## A.4 Execution Module

---

**Prompt for EXECUTION Module**

You are the controller of a highly trained agent within a grid forest world. Your job is to convert a single text action into a structured format for robotic control.

Here is the action we want to perform: [ACTION]

Your job is to convert the action into an executable format. Do not change the actions, just translate them.

This is the executable action format:

Action{
int "type": type of action being performed
int "param 1": parameter 1 of action if applicable
int "param 2": parameter 2 of action if applicable
string "description": description of action
}

You have X distinct types of actions. You MUST choose one of them:

1. Move to any coordinate location in one step regardless of distance:

"type": 1
"param 1": x coordinate of location
"param 2": y coordinate of location
"description": description of action

Example Action:
[1, 500, 500, "move to coordinate location of (500, 500)"]

[OTHER PRIMITIVES]

---

### A.5 Prompts of CAMON Generate Plan

---

**Prompt for CAMON: Generate Plan**

You are AGENT X, an AGENT-TYPE Agent, currently acting as the leader in a cooperative multi-agent robotic task. This is your team composition, (including yourself):

TEAM COMPOSITION
—

Your team's current task is:

CURRENT TASK
—

Your past actions were:

PAST ACTIONS
—

This is your chat history with agents in your team:

CHAT HISTORY
—

This is your team's (including you) collective observations, locations, current actions, and past actions of all agents.

GLOBAL DATA
—

Now your job is to provide the next best action for yourself, and OPTIONALLY: the next best action for any other agents. Remember, you are AGENT X an AGENT-TYPE Agent, located at POSITION.

These are all the possible actions for each type of agent. This is a comprehensive list, so the action MUST be one of these types. NO other responses are allowed.

LIST OF AGENT ABILITIES ON TEAM

Provide your output in the following format:

<reasoning>(any reasoning or calculations)</reasoning>

<action>'MY NEXT ACTION'</action>

OPTIONAL-for other agents:

<AGENT ID-action>(AGENT ID'S NEXT ACTION)<AGENT ID-action>
<AGENT ID-message>(message to AGENT ID)<AGENT ID-message>

For example:
<AGENT A-action>'action'<AGENT A-action>
<AGENT A-message>'action'<AGENT A-message>

---

### A.6 Prompts of CAMON Propose Plan

---

**Prompt for CAMON: Propose Plan**

You are AGENT X, an embodied AGENT-TYPE agent within a X by Y forest grid world and part of a collaborative team of N Agents.
This is your team's composition (including yourself):

TEAM COMPOSITION
—

These are your current observations:

PERCEPTION
—

This is your team's overall task:

CURRENT TASK
—

Your past actions were:

ACTION HISTORY
—

This is your chat history with agents in your team:

CHAT HISTORY
—

Your job is to propose your next action. These are your possible actions:

AGENT ABILITIES

This is a comprehensive list, so your action MUST be one of these types. NO other responses are allowed.

Provide your output in the following format:

<reasoning>(any reasoning or calculations)</reasoning>
<action>'MY NEXT ACTION'</action>

---

## A.7 Prompts of CAMON Review Plan

**Prompt for CAMON: Review Plan**

You are AGENT X, currently acting as the leader in a cooperative multi-agent robotic task.

This is your team composition (including yourself):

TEAM COMPOSITION
—

Your team's current task is:

CURRENT TASK
—

This is your teams'(including you) collective observations, locations, current actions, and past actions of all agents. Only you have all of this data.

GLOBAL DATA
—

Your teammate AGENT Y, an AGENT-TYPE Agent, is proposing a new action for itself:

PROPOSED ACTION
—

Your job is to review this action and ACCEPT or REJECT it.

Then provide the next best action for AGENT Y, choosing a better one if REJECT or repeating/rewriting the proposed one if ACCEPT. Also send a message to AGENT Y describing your choice.

Additionally, you may announce information to other agents in your team with information. You may also choose to override actions for other agents as well. You must send a message to that agent if you do so. This interrupts their action, so only do this if you want to change their current action.

These are all the possible actions for each type of agent. This is a comprehensive list, so the action MUST be one of these types. NO other responses are allowed.

LIST OF AGENT ABILITIES ON TEAM

Provide your output in the following format:
<reasoning>(any reasoning or calculations)</reasoning>
<decision> ACCEPT OR REJECT </decision>
<action> AGENT Y's next action </action>
<message> message to AGENT Y</message>

OPTIONAL-for other agents:
<AGENT ID-action>(AGENTID'S NEXT ACTION)<AGENT ID-action>
<AGENT ID-message>(message to AGENTID)<AGENT ID-message>

For example: <AGENT A-action>'action'</AGENT A-action>

### A.8 Prompts of COELA: Propose Message

---

**Prompt for COELA: Propose Message**

You are the communicator module of Agent X, a AGENT-TYPE Agent in a cooperative multi-agent robotic task.

This is your team composition, including you:

TEAM COMPOSITION

—

Your team's task is:
CURRENT TASK

—

Your status and observations:

PERCEPTION

—

Your chat history:

CHAT HISTORY

—

Your past actions:

ACTION HISTORY

—

Your job is to propose a message to send to the chat/groupchat.

Provide your output in the following format:

<reasoning>(any reasoning or calculations)</reasoning>

<message>'MESSAGE'</message>

Note: The generated message should be accurate, helpful, and brief. Do not generate repetitive messages

---

## A.9 Prompts of COELA: Choose Action

---

**Prompt for COELA: Choose Action**

You are Agent X, an AGENT-TYPE Agent in a cooperative multi-agent robotic task.

This is your team composition, including you:

TEAM COMPOSITION

—

Your team's task is:

CURRENT TASK

—

Your status and observations:

PERCEPTION

—

Your chat history:

CHAT HISTORY

—

Your past actions:

ACTION HISTORY

—

Now your job is to provide the next best action for yourself. Remember, you are Agent X an AGENT-TYPE Agent, located at POSITION.

These are all the possible actions for each type of agent. This is a comprehensive list, so the action MUST be one of these types. NO other responses are allowed. Note that sending messages has a cost so think about the necessity of it.

- [send message to groupchat] PPROPOSED MESSAGE
- OTHER ABILITIES

Provide your output in the following format:

<reasoning>(any reasoning or calculations)</reasoning>

<action>'MY NEXT ACTION'</action>

Include 'SEND MESSAGE' in all caps like so, if and only if your action is to send the message. For example:

<action>SEND MESSAGE 'proposed message'</action>

---

### A.10 Prompts of Embodied: Generate Communications

---

**Prompt for Embodied: Generate Communications**

You are AGENT X, a AGENT-TYPE Agent in a cooperative multi-agent robotic task.

Given your shared goal, chat history, and your progress and previous actions, please generate a list of short messages to members of your team in order to achieve the goal as possible.

This is your team composition, including you:

TEAM COMPOSITION

—

Your team's task is:

CURRENT TASK

—

Your status and observations:

PERCEPTION

—

Your past actions:

ACTION HISTORY

—

Your chats:

CHAT HISTORY

—

You may send messages to individual agents or in a global channel. Think about the necessity of sending a message. There are costs to send messages. Provide your output in the following format. All names should be in all caps:

<reasoning>(any reasoning or calculations)</reasoning>

<RECIPIENT>'MESSAGE'</RECIPIENT>
<GLOBAL>'MESSAGE'</GLOBAL>

For Example:

<AGENT A>message</AGENT A>,
<AGENT C>message</AGENT C>,
<GLOBAL>message</GLOBAL>

---

### A.11 Prompts of Embodied: Generate Actions

---

**Prompt for Embodied: Generate Action**

You are AGENT X, an AGENT-TYPE Agent in a cooperative multi-agent robotic task.

Your team's task is:

CURRENT TASK
—

Your status and observations:

PERCEPTION
—

Your chat history:

CHAT HISTORY
—

Your past actions:

ACTION HISTORY
—

Now your job is to provide the next best action for yourself. Remember, you are AGENT X an AGENT-TYPE Agent, located at POSITION.

These are all the possible actions for each type of agent. This is a comprehensive list, so the action MUST be ONE and only ONE of these types. NO other responses are allowed.

AGENT ABILITIES

Provide your output in the following format:

<reasoning>(any reasoning or calculations)</reasoning>

<action>'MY NEXT ACTION'</action>

Make sure you include enough details in your action such as explicit target coordinate locations. For example:

<action>Move towards (500,500)</action>

---

### A.12 Prompts of HMAS-2: Central Planner

---

**Prompt for HMAS-2: Central Planner**

You are central planner directing agents in a cooperative multi-agent robotic task.

Your team's task is:

CURRENT TASK

—

Your team's previous state action pairs at each step are:

STEP HISTORY

—

Your team's current state and available actions are:

GLOBAL STATE

—

Now your job is to provide the next best action for each agent. You must provide a single action for each agent. These actions must be exactly ONE of the agent's available actions, including the 'do nothing' action. Do not propose multiple actions per agent.

Specify your action plan in the following format with agent names in all caps:

<reasoning>(any reasoning or calculations)</reasoning>

<AGENT>'MY NEXT ACTION'</AGENT>

For example:

<AGENT A>'action'</AGENT A>
<AGENT B>'action'</AGENT B>

Make sure you include enough details in each action such as explicit target coordinate locations.

---

### A.13 Prompts of HMAS-2: Feedback

---

**Prompt for HMAS-2: Feedback**

You are AGENT X, an AGENT-TYPE Agent in a cooperative multi-agent robotic task.

Your team's task is:

CURRENT TASK

—

Your team's previous state action pairs at each step are:

STEP HISTORY

—

Your team's current state and available actions are:

GLOBAL STATE

—

The initial action plan from the central planner is:

INITIAL PLAN

—

Now your job is to provide feedback to the action plan specifically regarding your agent. If the plan is satisfactory, the feedback should only be 'ACCEPT'.

Remember, you are AGENT X an AGENT-TYPE Agent, located at POSITION.

<reasoning>(any reasoning or calculations)</reasoning>

<feedback>'feedback'</feedback>

---

### A.14 Baseline Pseudo Codes

#### A.14.1 CAMON Algorithm

---

**Algorithm 1:** CAMON Implementation

**Data:** Agent set $\mathcal{A}$, environment env, time limit $T_{\max}$, target score MaxScore, task task

**1** Leader $\leftarrow \mathcal{A}[0]$;
**2** state $\leftarrow$ env.reset();
**3 for** $t \leftarrow 1$ **to** $T_{max}$ **do**
**4**     GlobalData $\leftarrow \{\}$;
     /* Parsing Observations and Generating Perceptions             */
**5**     **foreach** $a \in \mathcal{A}$ **do**
**6**        $o_a \leftarrow$ GetObservation$_a$(state);
**7**        $p_a \leftarrow$ LLM_GeneratePerception$_a$($o_a$);
**8**        agentData$_a \leftarrow (p_a,$ action$_a, \mathcal{H}_a)$;
**9**        append agentData$_a$ to GlobalData;
     /* Generating Plan                                      */
**10**     **foreach** $a \in \mathcal{A}$ **do**
**11**        **if** $action_a \neq None$ **then**
**12**          **continue**;
**13**        **if** $a = Leader$ **then**
**14**          $plan \leftarrow$ LLM_GeneratePlan$_a$(task, $\mathcal{A}$, GlobalData, $\mathcal{M}_a$);
**15**        **else**
**16**          $proposal \leftarrow$ LLM_ProposePlan$_a$(task, $p_a, \mathcal{H}_a, \mathcal{M}_a$);
**17**          $plan \leftarrow$ LLM_LeaderResponse$_{\text{leader}}$(proposal, task, $\mathcal{M}_a$, GlobalData);
**18**          Leader $\leftarrow a$
       /* Assigning tasks to agents                        */
**19**        **foreach** $task_a \in plan$ **do**
**20**          action$_a \leftarrow$ LLM_TranslateAction(task$_a$);
**21**     $E \leftarrow \{\}$;
     /* Executing Actions                                   */
**22**     **foreach** $a \in \mathcal{A}$ **do**
**23**        $e_a \leftarrow$ ExecuteAction(action$_a$);
**24**        **if** *action is complete* **then**
**25**          append action$_a$ to $\mathcal{H}_a$;
**26**          $action_a =$ None;
**27**        $E[a] \leftarrow e_a$;
**28**     (state, score) $\leftarrow$ env.step($E$);
**29**     **if** $score \geq MaxScore$ **then**
**30**        **break**;

---

In our CAMON Implementation, a **leader agent** is initially chosen as shown in line 1. Then, for each timestep, each agent parses their observations($o_a$) from the environment states, and then generates a text perception ($p_a$) given those observations through the PERCEPTION module (A.2). This perception ($p_a$) is combined with the agent's current action (action$_a$) and its action history ($\mathcal{H}_a$) to create agentData$_a$, which is then appended to the GlobalData.

Then, each agent checks their current action (action$_a$). If it is still active (meaning it is multi-step and not complete), the agent will do nothing, as seen in line 12.

If not, and the agent is the **leader agent**, it will independently generate a new team plan through LLM_GeneratePlan$_a$ (A.5) in line 14. If the agent is not the **leader agent**, instead it will propose a

plan through $\texttt{LLM\_ProposePlan}_a$(A.6), in line 16. This plan will then be reviewed by the **leader agent** through $\texttt{LLM\_ReviewPlan}_{leader}$(A.7)in line 17. Then the agent will become the new **leader agent**.

Next, the finalized plan will be parsed into tasks for each agent and translated into a standardized action format through $\texttt{LLM\_TranslateAction}_a$ (A.4) in line 20. This action ($\texttt{action}_a$) will be each agent's active action.

Now since all agents have an active action, each agent will generate an executable vector ($e_a$) through our EXECUTION Module in line 23. If the current action ($\texttt{action}_a$) is complete or single-step, it will then be set to $\texttt{None}$ and appended to the agent's action history.

Lastly, all executable tensors will be combined and executed within the Wildfire Environment in line 28. If the max score has been reached, the algorithm ends.

### A.14.2 COELA Algorithm

---

**Algorithm 2:** COELA Implementation

---
**Data:** Agent set $\mathcal{A}$, environment env, time limit $T_{\max}$, target score MaxScore, task task

**1** $state \leftarrow$ env.reset();

**2** $\mathcal{M} \leftarrow \{\}$

**3** for $t \leftarrow 1$ to $T_{max}$ do

    /* Parsing Observations and Generating Perceptions                             */

**4**     foreach $a \in \mathcal{A}$ do

**5**         $o_a \leftarrow$ GetObservation$_a$(state);

**6**         $p_a \leftarrow$ LLM_GeneratePerception$_a$($o_a$);

    /* Proposing Message and Choosing Action                                    */

**7**     foreach $a \in \mathcal{A}$ do

**8**         if $action_a \neq None$ then

**9**             continue;

**10**         $proposedMessage \leftarrow$ LLM_ProposeMessage$_a$(task, $p_a$, $\mathcal{H}_a$, $\mathcal{M}$);

**11**         $chosenAction \leftarrow$ LLM_ChooseAction$_a$(task, proposedMessage, $p_a$, $\mathcal{H}_a$, $\mathcal{M}$);

**12**         if $chosenAction = proposedMessage$ then

**13**             append proposedMessage to $\mathcal{M}$

**14**             $action_a \leftarrow$ NoAction

**15**         else

**16**             $action_a \leftarrow$ LLM_TranslateAction(chosenAction);

**17**     $E \leftarrow \{\}$;

    /* Executing Actions                                                       */

**18**     foreach $a \in \mathcal{A}$ do

**19**         $e_a \leftarrow$ ExecuteAction($action_a$);

**20**         if *action is complete* then

**21**             append action$_a$ to $\mathcal{H}_a$;

**22**             $action_a =$ None;

**23**         $E[a] \leftarrow e_a$;

**24**     (state, score) $\leftarrow$ env.step($E$);

**25**     if $score \geq MaxScore$ then

**26**         break;

---

In our COELA Implementation, our message history ($\mathcal{M}$) is initially empty in line 2. Then, for each timestep, each agent parses their observations($o_a$) from the environment states, and then generates a text perception ($p_a$) given those observations through the PERCEPTION module (A.2).

Then, each agent checks their current action (action$_a$). If it is still active (meaning it is multi-step and not complete), the agent will do nothing, as seen in line 9.

However, if not, each agent proposes a message to send through LLM_ProposeMessage$_a$ (A.8) given the team's task, the agent's perception ($p_a$), the agent's action history ($\mathcal{H}_a$), and the team's chat history ($\mathcal{M}$).

Then, given the proposed message, through LLM_ChooseAction$_a$ (A.9), the agent chooses to execute an action or to send the proposed message.

If the agent chooses to send the proposed message, then that message is appended to the message history ($\mathcal{M}$) and No Action is chosen in lines 13-14.

However, if the agent chooses a different action, that action is translated into a standardized action format through LLM_TranslateAction$_a$ (A.4) in line 16. This action (action$_a$) will be each agent's active action.

Now since all agents have an active action (including `No Action`), each agent will generate an executable vector ($e_a$) through our EXECUTION Module in line 19. If the current action ($action_a$) is complete or single-step, it will then be set to `None` and appended to the agent's action history.

Lastly, all executable tensors will be combined and executed within the Wildfire Environment in line 24. If the max score has been reached, the algorithm ends.

### A.14.3 Embodied Algorithm

---

**Algorithm 3:** Embodied Implementation

**Data:** Agent set $\mathcal{A}$, environment `env`, time limit $T_{\max}$, task `task`, communication rounds $C$

1   $state \leftarrow$ env.reset();
2   **for** $t \leftarrow 1$ **to** $T_{max}$ **do**
     /* Parsing Observations and Generating Perceptions                        */
3     **foreach** $a \in \mathcal{A}$ **do**
4        $o_a \leftarrow$ GetObservation($a$, state);
5        $p_a \leftarrow$ LLM_GeneratePerception($a$, $o_a$);
     /* Communication Rounds                                         */
6     **for** $c \leftarrow 1$ **to** $C$ **do**
7        **foreach** $a \in \mathcal{A}$ **do**
8           $messages_a \leftarrow$ LLM_GenerateMessages(task, $p_a$, $\mathcal{H}_a$, $\mathcal{M}_a$);
9           **foreach** $(recipient, msg) \in messages_a$ **do**
10             append msg to $\mathcal{M}_a$;
11             append msg to $\mathcal{M}_{\text{recipient}}$;

     /* Generating Actions                                         */
12     **foreach** $a \in \mathcal{A}$ **do**
13        generatedAction $\leftarrow$ LLM_GenerateAction$_a$(task, $p_a$, $\mathcal{H}_a$, $\mathcal{M}_a$);
14        $action_a \leftarrow$ LLM_TranslateAction(generatedAction)
15     $E \leftarrow \{\}$;
     /* Executing Actions                                           */
16     **foreach** $a \in \mathcal{A}$ **do**
17        $e_a \leftarrow$ ExecuteAction($action_a$);
18        **if** *action is complete* **then**
19           append $action_a$ to $\mathcal{H}_a$;
20           $action_a =$ None;
21        $E[a] \leftarrow e_a$;
22     $(state, score) \leftarrow$ env.step(actions);
23     **if** *score $\geq$ MaxScore* **then**
24        **break**;

---

In our Embodied Implementation, we start each timestep with all agents parsing their observations ($o_a$) from the environment state in line 4, and then generating a text perception ($p_a$) given those observations through the PERCEPTION module (A.2) in line 5.

Then, for $c$ communication rounds, each agent generates messages for any recipient agent through LLM_GenerateMessages$_a$ (A.10) in line 8, given the team's task, the agent's perception, action history (($\mathcal{H}_a$), and message history ($\mathcal{M}_a$). These messages are then parsed and added to the corresponding agents' message histories (($\mathcal{M}_{\text{recipient}}$) in line 11.

Then after all communication rounds, each agent generates their next action with LLM_GenerateMessages$_a$ (A.11) in line 13, given the team's task, the agent's perception, action history ($\mathcal{H}_a$), and message history ($\mathcal{M}_a$). This generated action is then translated into a standardized action format through LLM_TranslateAction$_a$ (A.4) in line 14. This action ($action_a$) will be each agent's active action.

Now since all agents have an active action (including `No Action`), each agent will generate an executable vector ($e_a$) through our EXECUTION Module in line 17. If the current action ($action_a$) is complete or single-step, it will then be set to `None` and appended to the agent's action history.

Lastly, all executable tensors will be combined and executed within the Wildfire Environment in line 22. If the max score has been reached, the algorithm ends.

### A.14.4 HMAS-2 Algorithm

---

**Algorithm 4:** HMAS Implementation

**Data:** Agent set $\mathcal{A}$, environment `env`, time limit $T_{\max}$, task `task`

**1** Leader $\leftarrow \mathcal{A}[0]$;
**2** GlobalState $\leftarrow \{\}$;
**3** StepHistory $\leftarrow \{\}$;
**4** state $\leftarrow$ env.reset();
**5 for** $t \leftarrow 1$ **to** $T_{max}$ **do**

    /* Parsing Observations and Generating Perceptions    */
**6**    **foreach** $a \in \mathcal{A}$ **do**
**7**        $o_a \leftarrow$ GetObservation($a$, state);
**8**        $p_a \leftarrow$ LLM_GeneratePerception($a$, $o_a$);
**9**        append $p_a$ to GlobalState[$a$];

    /* Iterating Plans    */
**10**    valid $\leftarrow$ False
**11**    **while** $valid = False$ **do**
**12**        valid $\leftarrow$ True
**13**        plan $\leftarrow$ LLM_GeneratePlan$_{leader}$(task, GlobalState, StepHistory, Review);
        /* Reviewing Plan    */
**14**        Review $= \{\}$
**15**        **foreach** $a \in \mathcal{A}$ **do**
**16**            feedback$_a \leftarrow$ LLM_ReviewPlan$_a$(plan, GlobalState, Step History);
**17**            **if** $feedback_a$ $is$ $REJECT$ **then**
**18**                append feedback$_a$ to Review;
**19**                valid $\leftarrow$ False;

**20**    **foreach** $task_a \in plan$ **do**
**21**        action$_a \leftarrow$ LLM_TranslateAction(task$_a$)

    /* Executing Actions    */
**22**    actions $\leftarrow \{\}$;
**23**    **foreach** $a \in \mathcal{A}$ **do**
**24**        $e_a \leftarrow$ ExecuteAction(action$_a$);
**25**        **if** $action$ $is$ $complete$ **then**
**26**            append action$_a$ to $\mathcal{H}_a$;
**27**            action$_a =$ None;
**28**        $E[a] \leftarrow e_a$;

**29**    append (GlobalState, actions) to StepHistory;
**30**    (state, score) $\leftarrow$ env.step(actions);
**31**    **if** $score \geq MaxScore$ **then**
**32**        **break**;

---

In our HMAS-2 Implementation, a **leader agent** initially is chosen shown in line 1. Then, for each timestep, each agent parses their observations($o_a$) from the environment states, and then generates a text perception ($p_a$) given those observations through the PERCEPTION module (A.2). This perception($p_a$) is then appended to the GlobalState.

Then, the **leader agent** generates a plan for the team through LLM_GeneratePlan$_{leader}$ (A.12) given the team's task, the Global State, the Step History, and reviews from other agents in line 13. This will provide the leader agent with all agents' current perceptions and past perceptions and actions.

Then each agent will review the plan and provide feedback through $\texttt{LLM\_ReviewPlan}_a$ (A.13) in line 16. If the feedback is to reject the plan, the feedback will be appended to $\texttt{Review}$ and $\texttt{valid}$ will be set to $\texttt{False}$. This will cause lines 12-19 to be repeated, reiterating the plan.

When the plan does not have any rejection, the finalized plan will be parsed into tasks for each agent and translated into a standardized action format through $\texttt{LLM\_TranslateAction}_a$ (A.4) in line 20. This action ($\texttt{action}_a$) will be each agent's active action.

Now, since all agents have an active action, each agent will generate an executable vector ($e_a$) through our EXECUTION Module in line 23. If the current action ($\texttt{action}_a$) is complete or single-step, it will then be set to $\texttt{None}$ and appended to the agent's action history.

Lastly, all executable tensors will be combined and executed within the Wildfire Environment in line 28. If the max score has been reached, the algorithm ends.

## A.15 VLM vs. Perception Module Ablation Study

We conducted an evaluation to assess the accuracy of our perception module versus the GPT4o VLM on 100 sample observations. Given the same game frames, the perception module received the associated minimap ASCII representation while the VLM received the image directly. Both were prompted by the same perception prompt (see Appendix A.2), with slight adjustments where the VLM was given a color key instead of an ASCII key.

An unbiased JudgeLLM (GPT4o) judged both perception summaries against the ground truth data with respect to accuracy in locating features such as fires and civilians. The prompt for the Judge LLM is as follows:

> **JudgeLLM Evaluation Prompt**
>
> Ground Truth Data (extracted from the environment): {ground_truth_json}
> Perception Text to Evaluate: "{perception_text}"
> Evaluate this perception text and provide:
>
> 1. Fire Detection Score (0-10): How accurately did it identify fires (ignited, on fire, extinguishing cells)?
>
> 2. Civilian Detection Score (0-10): How accurately did it identify civilians?
>
> Respond using this format:
> <fire_score>NUMBER</fire_score> <civilian_score>NUMBER</civilian_score> <explanation> Brief explanation of your scoring </explanation>

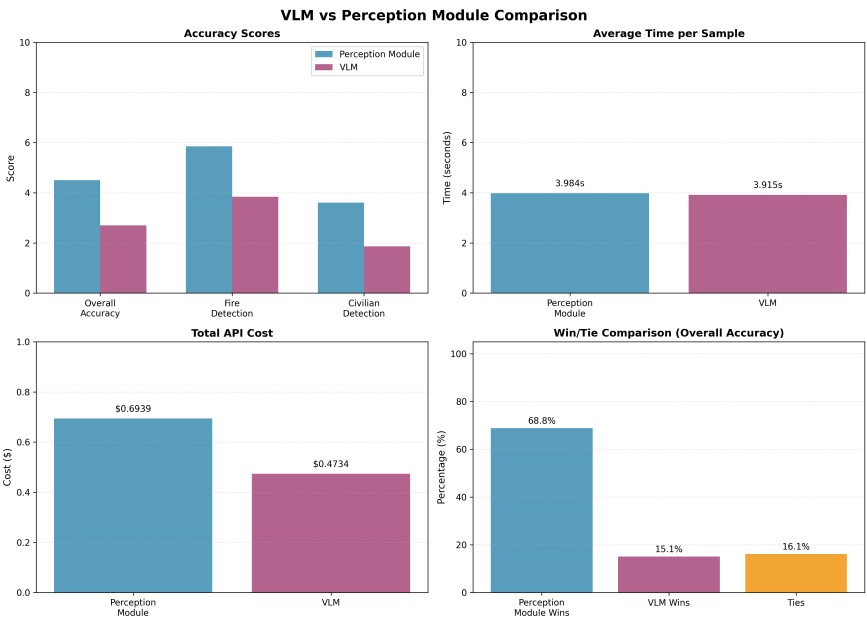

Figure 9: **Behavior Competency Scores (BCS) by algorithm.**

These figures demonstrate that the perception module is more accurate in our setting and does not significantly lag in terms of speed or cost.

### A.16 Computational Performance

Table 6: Resource usage with 20 agents across different map sizes.

| Map Length (cells) | CPU Usage (%) | RAM Usage |
|---|---|---|
| 250 (62,500) | 27.2 | 206 MB |
| 400 (160,000) | 30.1 | 262 MB |
| 600 (360,000) | 25.5 | 353 MB |
| 800 (640,000) | 26.8 | 482 MB |
| 1000 (1,000,000) | 36.4 | 604 MB |

Table 7: Resource usage with different agent counts on a 100-size map.

| Agent Count | CPU Usage (%) | RAM Usage |
|---|---|---|
| 20 Agents | 25.4 | 182 MB |
| 100 Agents | 27.5 | 266 MB |
| 500 Agents | 20.0 | 1034 MB |
| 1000 Agents | 21.6 | 1990 MB |
| 2000 Agents | 25.3 | 3835 MB |

### A.17 Emergent behaviors observed in dialogues

Table 8: **Examples each CREW-Wildfire Behavioral Goals**

| Behavioral Goal | Explanations | Examples of dialogue snippets that reflect such behaviors |
|---|---|---|
| Task Designation (TD) | Agents explicitly split tree-cutting targets so no effort overlaps. | We should divide the target locations... 
 – AGENT_1 → (11, 16) to cut trees. 
 – AGENT_2 → (13, 16) to cut trees. 
 – AGENT_3 → (11, 3) to cut trees. |
| Agent Capitalization (AC) | Firefighter delegates tree-line task to Bulldozer with superior cutting. | AGENT_1: "Since you have exceptional tree-cutting abilities, could you focus on the line from (26, 6) to (26, 10)? AGENT_2 and I will support you as needed." 
 AGENT_3: "Confirmed, I will focus on cutting trees from (26, 6) to (26, 10). Currently moving towards the target area." |
| Spatial Reasoning (SR) | Uses relative offsets to locate and frame the fire zone spatially. | "AGENT_3 at (130, 200). Fire roughly 12 E / 11 S. Requesting confirmation assistance." |
| Observation Sharing (OS) | Broadcasts civilian position and hazard status for rescue action. | "AGENT_5: Four civilians at (230, 124). No fires. Request Firefighters to escort to safe zone." |
| Realtime Coordination (RC) | Synchronizes boarding with helicopter arrival. | AGENT_1: "All agents, prepare for transport by AGENT_7. We will move towards the target location [92.0, 11.0]" 
 AGENT_5: "Ready for transport. AGENT_7, please initiate pick-up and transport operation as planned" 
 AGENT_7: "Initiating pick-up and transport operation" 
 AGENT_7: "Confirming all Firefighter Agents are on board. Departing now" |
| Plan Adaptation (PA) | Detects new fire spread; requests aerial support, altering strategy. | "AGENT_12: Fire spreading north at (238, 198). Need aerial backup from Helicopter 13 ASAP." |
| Objective Prioritization (OP) | Balances search mission with stand-by evacuation planning. | "AGENT_2 now searching for civilians ahead of fire at (520, 565). Helicopter 10, stay ready to assist evacuation." |

### A.17.1 Example Calculation of BCS for CAMON (`algorithm`) and `Realtime Communication` (`behavioral goal`)

| Level $\ell$ | Raw score $s_{a,\ell}$ | Target $T_\ell$ | Baseline $B_\ell$ | $NS_{a,\ell}$ |
|---|---|---|---|---|
| Transport FF (S) | 6.00 | 6 | 0 | 1.000 |
| Transport FF (L) | 10.00 | 12 | 0 | 0.833 |
| Rescue $S + R + T$ | 0.00 | 10 | 0 | 0.000 |
| Suppress $L + T + S$ | $-729.67$ | 0 | $-929.67$ | 0.281 |
| Full Environment | $-5571.67$ | 0 | $-5722.67$ | 0.038 |

$$\text{BCS}_{\text{CAMON,RC}} = \frac{1.000 + 0.833 + 0.000 + 0.281 + 0.038}{5} \approx \mathbf{0.430}.$$

### A.18 Behavior Competency Scores (BCS) Radar Chart

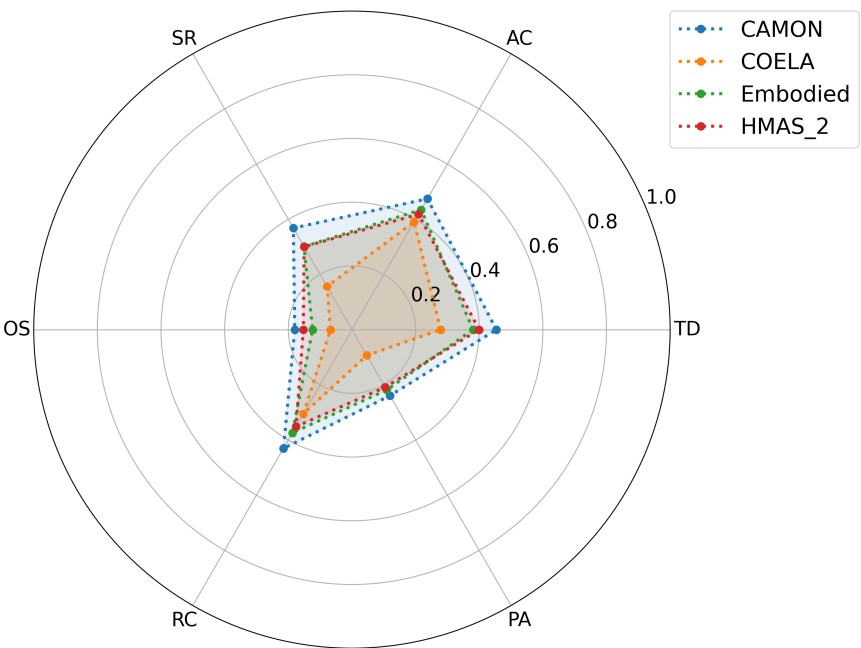

Figure 10: **Behavior Competency Scores (BCS) by algorithm.**

**A.19   Random Seeds and Other Hyperparameters**

For all the LLMs, we set the temperature to 0 to achieve deterministic results and a single completion per decision step. All experiments used GPT-4o (model ID: gpt-4o-2024-08-06) with a context window of 128,000 tokens.

Baseline-specific hyperparameters were set as follows:

- **Embodied Implementation:**
    - Communication Rounds per Timestep: 2
    - Message Lifespan: 3 timesteps

- **COELA:**
    - Max Messages: 30 messages per chat history

Note: COELA uses a maximum message count while Embodied uses message lifespan because COELA maintains a single continuous chat while Embodied uses independent chats that are synchronized.

The generation seeds for the tasks are listed in Tab. 9.

Table 9: **Seeds Used Across All Levels**

| Level | Seeds Used |
|---|---|
| Cut Trees: Sparse (Small) | 375, 483, 43, 6370, 9964, 2097, 25808, 83248, 48320, 94510 |
| Cut Trees: Sparse (Large) | 212, 981, 1530, 5382, 9405 |
| Cut Trees: Lines (Small) | 9259, 4881, 8456, 59497, 66768, 78914 |
| Cut Trees: Lines (Large) | 820, 5406, 6503 |
| Scout Fire (Small) | 4651, 6841, 7593, 1012, 8528, 29751, 36346, 42589, 43846, 62563 |
| Scout Fire (Large) | 5324, 3603, 8592, 43126, 70576 |
| Transport Firefighters (Small) | 283, 2461, 2478, 7622, 7647 |
| Transport Firefighters (Large) | 741, 7305, 9528, 8079, 6232 |
| Rescue Civilians: Known Location (Small) | 9502, 3972, 6545, 5884, 8491, 7723, 30743, 51358 |
| Rescue Civilians: Known Location (Large) | 7979, 1539, 2269, 7152, 5226 |
| Rescue Civilians: Search and Rescue | 966, 7377, 7285 |
| Rescue Civilians: Search + Rescue + Transport | 8208, 150, 2577 |
| Suppress Fire: Extinguish | 2994, 4936, 4847 |
| Suppress Fire: Contain | 733, 7765, 8049 |
| Suppress Fire: Locate and Suppress | 5280, 2142, 2628 |
| Suppress Fire: Locate + Transport + Suppress | 6309, 3821, 6117 |
| Full Environment | 6434, 9424, 9500 |

