# OpenReview forum: "CREW-Wildfire: Benchmarking Agentic Multi-Agent Collaborations at Scale"
_TMLR — Accepted by TMLR_

### Review · Reviewer_6WkG · 2025-10-13

**Summary Of Contributions:**

This paper introduces CREW-Wildfire, a benchmark environment for evaluating LLM-based multi-agent systems in complex, large-scale wildfire response scenarios. The authors argue that existing benchmarks lack the scale, complexity, and realism needed to properly test multi-agent coordination capabilities. Built on the CREW human-AI teaming platform, CREW-Wildfire features procedurally generated environments with up to 2000+ agents, heterogeneous roles (drones, helicopters, bulldozers, firefighters), partial observability, and stochastic dynamics. The paper evaluates four LLM-based agent frameworks (AutoGen, CAMON, LangGraph, CrewAI) across 12 benchmark levels measuring 7 behavioral competencies. Results show current frameworks struggle significantly with spatial reasoning, real-time coordination, and plan adaptation.

**Strengths**

- The benchmark addresses a limitation in current multi-agent evaluation with most existing work focusing on small-scale, fully observable, or simplified environments. The wildfire domain serves as a prototypical complex coordination task. In particular, supporting 2000+ agents substantially exceeds existing benchmarks.
- The 7 behavioral competencies (Task Designation, Agent Capitalization, Spatial Reasoning, etc.) provide interesting metrics beyond just task success and the Behavior Competency Score (BCS) methodology allows diagnosing specific failure modes.
- The results clearly demonstrate current LLM frameworks fail dramatically on complex coordination tasks which is valuable negative evidence for the community.

**Weaknesses**

- Perception module ablations: It would have been good to provide ablations for the specific implementation of the perception module. For example, using a strong VLM to map the image data to text instead of encoding it into ASCII first.
- BCS methodology: The target and baseline definitions seem somewhat arbitrary. The authors do not provide experiments to check how sensitive the BCS scores to their particular choices.

**Additional Comments:**

- What are the numbers in Table 3 showing?
- What happens if you give agents perfect communication or full observability? How much does partial observability matter?
- How does performance decrease with the number of agents?
- What makes "Large" harder than "Small". Is it just map size or other factors? How was difficulty validated?
- The "Do Nothing" baseline is useful but extremely weak. Would it be possible to try a scripted heuristic baseline? That could be more informative.

**Audience:**

Yes

**Audience Explanation:**

This paper tackles an important problem and introduces a technically impressive benchmark that could be valuable for the community.

**Broader Impact Concerns:**

No concerns

**Claims And Evidence:**

Yes

**Claims Explanation:**

The paper’s claims are well grounded in the experimental evidence provided by the authors.

**Requested Changes:**

The following changes could strengthen the paper:

- Analyze the agent communication and perform a more detailed trajectory analysis showing patterns beyond individual examples. While the authors show examples of failure cases, it would be interesting to see statistics and a more high-level analysis as well. E.g., how often does a certain failure case happen?
- Provide a VLM baseline. The authors use GPT4o for their experiments already. This model also has has vision capabilities. It could be interesting to feed the image to the model directly instead of converting it to ASCII first.
- The paper focuses on task success but doesn't measure communication efficiency, redundancy, or other coordination-specific metrics beyond the behavioral goals.

---

> ### Author Response · Authors · 2025-10-25
> **Response to Reviewer 6WkG**
>
> We thank the reviewer for the thoughtful comments. We would like to address all of your concerns and questions below with point responses.
>
>
> ## Point 1: Perception Module with VLM
>
> *“Perception module ablations: It would have been good to provide ablations for the specific implementation of the perception module. For example, using a strong VLM to map the image data to text instead of encoding it into ASCII first”*
>
> *“Provide a VLM baseline. The authors use GPT4o for their experiments already. This model also has has vision capabilities. It could be interesting to feed the image to the model directly instead of converting it to ASCII first.”*
>
> We are happy to share that we have conducted an additional evaluation to assess the accuracy of our perception module versus the GPT4o VLM on 100 sample observations. Given the same game frames, the perception module was given the associated minimap ASCII representation while the VLM was given the associated image directly. Both were prompted by the same perception prompt listed in the Appendix, with slight adjustments where the VLM was given a color key instead of an ASCII key. Then, an unbiased JudgeLLM (GPT4o) judged both perception summaries against the ground truth data with respect to accuracy in locating features such as fires and civilians.
>
> The prompt for the Judge LLM is as follows:
>
> ```
> Ground Truth Data (extracted from the environment):
> {ground_truth_json}
>
> Perception Text to Evaluate:
> "{perception_text}"
>
> Evaluate this perception text and provide:
> 1. Fire Detection Score (0-10): How accurately did it identify fires (ignited, on fire, extinguishing cells)?
> 2. Civilian Detection Score (0-10): How accurately did it identify civilians?
>
> Respond using this format:
>
> <fire_score>NUMBER</fire_score>
> <civilian_score>NUMBER</civilian_score>
> <explanation>
> Brief explanation of your scoring
> </explanation>
> ```
>
> The results of these were tallied up and are presented here in these figures:
> **https://postimg.cc/bS28QZfd**
>
> These figures demonstrate that the perception module is more accurate in our setting and does not significantly lag in terms of speed or cost. We will include this ablation study in our revised paper.
>
>
> ## Point 2: BCS Calculation
>
> *“BCS methodology: The target and baseline definitions seem somewhat arbitrary. The authors do not provide experiments to check how sensitive the BCS scores to their particular choices”*
>
> We would like to clarify that our BCS baseline and target scores are based on objective values:
> the worst possible score on a task (doing nothing for finite tasks and losing all agents for open-ended tasks) and the hypothetical best possible score on a task (max reward for finite tasks and 0 for open-ended tasks). All BCS calculations are based on these scores only.
>
>
> ## Point 3: Communication Metrics
>
> *“The paper focuses on task success but doesn't measure communication efficiency, redundancy, or other coordination-specific metrics beyond the behavioral goals.”*
>
> *“Analyze the agent communication and perform a more detailed trajectory analysis showing patterns beyond individual examples. While the authors show examples of failure cases, it would be interesting to see statistics and a more high-level analysis as well. E.g., how often does a certain failure case happen?”*
>
> Thank you for this insightful feedback. We fully agree that this is an important problem of locating where and how failures occur in complex long-horizon tasks with open-ended communication. However, we believe this is a much larger and more complex problem to effectively and thoroughly analyze these factors. In tasks as long and complex as ours, catastrophic mistakes do not occur in simple errors, but rather are propagated through iterations of nuanced misjudgments, making them very difficult to locate. For example, a poor decision to scout for the fire before civilians can only be debated as a mistake after further misjudgments lead to their destruction. Moreover, the very infrastructure of diverse multi-agent frameworks, as shown in our four baselines, may inherently lead to different types of errors, making them incomparable. Manually detecting these errors also does not scale. We hypothesize that more complex, systematic, and quantitative methods need to be developed to measure communication efficiency, redundancy, and other coordination-specific metrics for large-scale mult-agent Agentic AI problems. To our best knowledge, this still remains an open research problem.
>
> We do plan to tackle this important problem in its own future work; However, it is beyond the scope of our current paper. However, we will open-source all recorded observations, actions, and chats of all agents, allowing for deeper analysis in the future. We will note this exciting future opportunity in our revised paper as future work.

---

> > ### Comment · Reviewer_6WkG · 2025-11-15
> > **Acknowledging response**
> >
> > Thanks a lot for these clarifications. I will consider them for my final recommendation.

---

> > > ### Author Response · Authors · 2025-11-15
> > > **Response**
> > >
> > > Thank you for your reviews and considerations. Please feel free to let us know if you have further questions.

---

> ### Author Response · Authors · 2025-10-25
> **Response to Reviewer 6WkG cont.**
>
> ## Miscellaneous Questions
>
> Thank you for these valuable questions. We will make these clear in our revised paper.
>
> *“What are the numbers in Table 3 showing?”*
>
> Table 3 presents the scores across for each state-of-the-art baseline on the 12-level benchmark. The scores for each level are calculated based on each level’s objective/scoring function. These scoring functions can be found in Table 2. We have also bolded the best-scoring baseline for each level.
>
> *“What happens if you give agents perfect communication or full observability? How much does partial observability matter?”*
>
> For each of our baselines, we allowed free communication with no environmental communication restraints, such as delay or range. Communication was instead defined by how each baseline implemented it in their original work, whether in consistent communication rounds or conditional handoffs.
>
> On the other hand, full observability is inherently infeasible. The input tokens required to handle maps that can take up to 200x200=40000 cells, exceed practicality in terms of context windows, time, and cost in current LLMs. Moreover, partial observations and imperfect communications reflect real-world challenges. We believe that inherent partial observability is a strength of our benchmark, demonstrating how real-world problems are usually not fully observable due to size and resource constraints.
>
> *“How does performance decrease with the number of agents?”*
>
> Table 3 shows our different runs with various levels, such as “Cut Trees: Sparse” at different agent scales. We noticed a corresponding drop in performance compared to possible scores as the agent counts grew. Moreover, overall across all levels, those with higher agent counts performed worse than those with fewer. However, this question points out the difficulty of noticing this unless one combines the results in Table 3 with the maximum scores of Table 2 to find how the relative performance drops. We will adjust Table 3 to include the maximum scores for each level to make this more evident. Thank you for this feedback.
>
> *“What makes "Large" harder than "Small". Is it just map size or other factors? How was difficulty validated?”*
>
> The differences between Large and Small variants are: map size, agent count, and some objectives (such as how many trees to cut). We will explicitly add this to the revised paper to make this clear. For difficulty, given that the relative scores decrease on larger variants, as per the previous question, we can conclude that they are more difficult. However, across different levels, such as “Scout Fire” with “Transport Firefighters,” we intentionally do not label one as more difficult than the other since they are not directly comparable, and difficulty cannot be validated.
>
> *“The "Do Nothing" baseline is useful but extremely weak. Would it be possible to try a scripted heuristic baseline? That could be more informative.”*
>
> We agree that “Do Nothing” is a weak baseline. However, while a heuristic baseline might be possible for simple tasks, it is infeasible for longer, larger, and more complex tasks with heterogeneous interdependent agents. Moreover, although a baseline for a “decent” policy would be useful, we would like to stress that this paper is exactly that benchmark. Our four baselines in the paper are state-of-the-art methods for LLM-based multi-agent Agentic AI frameworks. These baselines are to show the latest progress in the community and call for further developments on challenging and complex tasks. We intend for these baselines, instead of just the “Do Nothing” baseline, to serve as the benchmark for what the current state-of-the-art can perform.

---

### Review · Reviewer_qfL5 · 2025-10-17

**Summary Of Contributions:**

This paper introduces CREW-wildfire, a simulator for wildfire responses, as a multi-agent agentic (in the LLM-sense) environment. This environment requires detecting fires, searching for survivors, rescue, and managing fires. The system is made to be compatible with LLMs, by translating the environment details into language for use by language models to make decisions which can be translated into actions in the environment. This environment itself demands capabilities that we would want in our agents. These include large maps, heterogeneous agents, partial observability, stochastic dynamics, and long-horizon planning objectives, as stated in the paper. There are four kinds of heterogeneous  agents: firefighters, bulldozers, drones, and helicopters. It supposedly offers twelve levels (though table 2 and 9 appear to show 16 levels), that represent differents kinds of tasks in different forms. The paper tests several frameworks on these 12 (or 16?) tasks. It benchmarks their scores on all the tasks/levels and additionally produces a behavior-competency score to provide insight on the behavioral properties of different algorithms.

**Audience:**

Yes

**Audience Explanation:**

This paper introduces the CREW Wildfire environment, which in itself certainly does seem to offer something different and useful to a community interested in multi-agent agentic collaboration. The problem setting itself is of interest, and indeed, by my assessment, can be used to investigate research questions.

**Broader Impact Concerns:**

While this is open-ended with the use of language models, this work is firmly a research project. I do not foresee any issues with broader impact.

**Claims And Evidence:**

No

**Claims Explanation:**

I will begin with claims that are indeed validated.

> A new open-source benchmark for evaluating LLM-based multi-agent Agentic systems in procedurally generated, physically grounded, and high-stakes disaster response environments.

The paper indeed does this.

>CREW-Wildfire bridges these gaps by offering a fully open-source, embodied, and highly scalable environment centered on evaluating and benchmarking LLM-based multi-agent Agentic AI frameworks. It supports
heterogeneous agents, complex and dynamic terrain, and realistic objectives under partial observability and
stochastic conditions. By combining these features with support for low-level control and high-level language
reasoning, CREW-Wildfire provides a unique testbed for evaluating large-scale collaboration, perception,
and planning in Agentic multi-agent systems.

Roughly speaking, I do think this paper mostly achieves this.

> We implement and evaluate several state-of-the-art LLM-based multi-agent Agentic AI frameworks, uncovering significant performance gaps that highlight the unsolved challenges in large-scale coordination, communication, spatial reasoning, and long-horizon
planning under uncertainty

This paper introduces an environment that is argued demonstrates limitations in existing methods.
One weakness of this paper, in terms of a lack of convincing evidence, is a clear signal that the environment indeed demands these skills.
Notably, this paper lacks any evidence or demonstration that this task can be handled appropriately. For example, as the system is set up, it seems that minimal effort would be required to produce a human baseline for this task, if support is made for the low-level interface. Particularly since many of the tasks do not require several agents. At the moment, we have to take it somewhat on faith that agents with certain competencies would indeed perform well in this environment.


One thing that is not clear to me, are the mechanisms of communication. One of the stated contributions has to do with communication. I realize some of the baselines have leaders that can communicate to other agents. What is not clear from the text of the paper, however, is how the agents communicate. Figure 4 lists out the abilities of the different types of agents, none of which involve actual communication. It does beg the question then, how the different entities are meant to communicate in a manner that is semantically consistent with wildfire responses. For example, are the agents allowed to communicate with each other on the algorithm-side (as opposed to communicating through the environment) as per the pseudocode? Is this meant to simulate radio communication?

> To address this, CREW-Wildfire Benchmarking Suite provides infrastructure for systematic evaluation, testing, and
behavioral analysis

The paper lists what the behavioral goals are, but in the paper (including table 8), there is no enumeration of which behavior goals correspond to which tasks/levels. Moreover, the behavior goal designations are entirely qualitative in nature. Unfortunately, I have no means to evaluate the validity or correctness of the (unlisted) behavior goal labels with respect to the tasks.

The behavioral competency score is also hard to validate, and is unconvincing as evidence. It seems that each level or task has potentially multiple behavioral goals assigned to it. However, when computing the BCS, there is no clear way to decouple the degree to which each task requires competency in each behavioral goal. This can lead to a large amount of correlation that makes it extremely difficult to know, or even believe, whether the BCS is capturing the behavior meant to be measured.

As a side note, it is not clear the maximum score of 0 is indeed accurate for open-ended tasks. Is it possible to achieve a score of 0 on the open-ended tasks with negative penalties?


Other general issues:
- Only 3 seeds are run for everything, which definitely hurts the convincingness of the results.


Overall, I think this could be an effective environment for the development and evaluation of multi-agent agentic systems. Much of my concern has to do with my ability to verify the efficacy given what I have in the paper. A large part of this paper is that it offers an environment meant to test capabilities not offered by other environments, which indeed it seems to do. However, where this paper seems to fall short in terms of convincingness is its ability to clearly demonstrate those capabilities (which are largely qualitative with no strong validated measurements or proxies) are captured by the environment as well as demonstrating the inefficacy of baselines on said capabilities.

**Requested Changes:**

Critical Adjustments:
- It appears like there are 16 levels in tables 2 and 9. Clarifying whether there are 12 or 16 levels is important, as the entire paper's empirical results hinge on this.


Important adjustments:
- The pseudocode seems to suggest that the baselines do not have hyperparameters. Clarifying whether this is the case is helpful for understanding.
- The costs of running an environment are naturally of critical importance for the community and readers to understand whether the environment should be used. Figure 7 does not make it clear how many API calls come from the baseline methods versus actually making the API calls for running the environment itself. Clarification here is key.
- I am looking at the pseudocode for HMAS, Embodied Implementation, COELA, and CAMON. It does not appear that they have any learning. If there is no learning, I think it is important clarify in the text.


Helpful adjustments:
- Should use \citep (instead of cite or citet). For example, the entire related work section has citations in text that is grammatically incorrect. These should be parenthetical or numbered citations (whatever is required by the TMLR format)
- Fig. 7 is referenced, seemingly out of place, in section 4. Resolving this would benefit the clarity.
- "A.18 Random Seeds and Other Hyperparameters" (no hyperparameters are mentioned?)
- A.17: "compentency" should be "competency"
- In the execution model prompt on page 20, "PRIMATIVES" should be "primitives"

---

> ### Author Response · Authors · 2025-10-25
> **Response to Reviewer qfL5**
>
> We thank the reviewer for the thoughtful comments. We would like to address all of your concerns and questions below with point responses.
>
> ## Point 1: Lack of evidence that the environment can be demonstrated appropriately
> *“This paper introduces an environment that is argued demonstrates limitations in existing methods. One weakness of this paper, in terms of a lack of convincing evidence, is a clear signal that the environment indeed demands these skills. Notably, this paper lacks any evidence or demonstration that this task can be handled appropriately.”*
>
> This is a very valid observation given that, as you noticed, none of the state-of-the-art baselines were indeed able to handle the task appropriately. Even CAMON, which performed most consistently on simple levels, was unable to make any progress in the most complex task. However, this does not invalidate our claims that our environment features challenges not found in other existing work, such as complex long-horizon tasks, partial observability, and heterogeneous agents, nor does it invalidate our claim that it demands higher-level skills such as real-time coordination and plan adaptation. For example, regardless of whether any featured baseline experimentally performed real-time coordination, the “Transport Firefighters” task inherently requires it because of the pick-up and drop-off mechanic.
>
> *“For example, as the system is set up, it seems that minimal effort would be required to produce a human baseline for this task, if support is made for the low-level interface. Particularly since many of the tasks do not require several agents. At the moment, we have to take it somewhat on faith that agents with certain competencies would indeed perform well in this environment.”*
>
> We agree that a human baseline would be interesting to analyze, and our environment does already support low-level control. However, the observation that “many of the tasks do not require several agents” is incorrect, as all levels require multiple agents, with some requiring more than 10. As such, studying human control is not as simple as benchmarking one human acting as a human policy, but rather it must study complex human-teaming in large dynamic tasks, which is out of the scope of this project. However, we plan to study this essential idea in future work with CREW-Wildfire as we explicitly explore human-ai teaming at scale. We will mention this in our future work in the revised paper. Moreover, we will open source the entire project, including our dataset and codebase, to support future explorations on human baselines.
>
> ## Point 2: Mechanisms of Communication
>
> *“One thing that is not clear to me, are the mechanisms of communication. One of the stated contributions has to do with communication. I realize some of the baselines have leaders that can communicate to other agents. What is not clear from the text of the paper, however, is how the agents communicate.”*
>
> *“For example, are the agents allowed to communicate with each other on the algorithm-side (as opposed to communicating through the environment) as per the pseudocode?”*
>
>
> Yes, agents communicate through the algorithm side. However, as you noted, the methods of exactly how they communicate vary between baselines based on their original algorithm designs. Some may have free, open-ended communication where any agent can send messages to any other agents, while others may have directional communication or leader agents that can constrain communication between certain agents. Interestingly, our results have shown that the more freedom agents have in communication, it does not necessarily correlate with better teamwork. On the technical side, these messages work by prompting agents to write messages to other agents, followed by the constraints posed by each baseline algorithm, which are then aggregated into the recipient agent’s prompt. For example, see example at A.10 on how the Embodied baseline does it.

---

> > ### Comment · Reviewer_qfL5 · 2025-11-02
> > **Response to Points 1 and 2**
> >
> > Thanks for engaging with the feedback.
> >
> > ## Point 1
> > Your response to point 1 makes sense. But it will be beneficial to be more clear in mapping the statements of "our environment requires X capability" to the aspect of the environment that indeed does this. This is not strictly necessary everywhere --- for example the point that the environment involves heterogeneous agents is obvious given the listing out of the agents. However, for things like "long-term planning" I don't recall any explicit pointers (please correct me if I'm wrong) on what makes this long-horizon relative to others. It seems we are to just look at the task and infer this.
> >
> > However, as you did in your response, pointing out how real-time coordination is demanded in the pick-up and drop-off of firefighters was an excellent.
> >
> >
> > ## Point 2
> > If the the communication is algorithm-side, do you mind responding to my point about semantic consistency with wildfire responses?

---

> > > ### Author Response · Authors · 2025-11-06
> > > **Response to Points 1 and 2**
> > >
> > > Thank you so much for your considerate responses. You emphasize a fair number of excellent points that we would like to further address.
> > >
> > > ## Point 1: Lack of evidence that the environment can be demonstrated appropriately
> > >
> > > *“Your response to point 1 makes sense. But it will be beneficial to be more clear in mapping the statements of "our environment requires X capability" to the aspect of the environment that indeed does this. This is not strictly necessary everywhere --- for example the point that the environment involves heterogeneous agents is obvious given the listing out of the agents. However, for things like "long-term planning" I don't recall any explicit pointers (please correct me if I'm wrong) on what makes this long-horizon relative to others. It seems we are to just look at the task and infer this.”*
> > >
> > > We will gladly add explicit pointers defending each of our claims about our environment to add clarity. We plan to add a section in the Appendix and point to it under this claim as follows:
> > >
> > > **Long-term complex planning:** Our benchmark's 'full game' level extends to 300 decision steps with a total runtime of around 20 minutes. This supports multiple possible subplans all within a single trajectory, including scouting for fires, rescuing civilians, finding water sources, and suppressing fires. Compared to traditional multi-agent environments like Petting Zoo or BoxNet1, where tasks are as simple as Atari games or moving boxes to target locations, our task suite is significantly more complex and long-horizon. Even the more sophisticated frameworks that use Habitat Sim involve relatively simple tasks like finding a given. In our environment, since critical information, such as fire locations and civilian positions, is unknown at the start, task planning must be high-level and adaptable. Complete upfront planning with simple execution is impossible in our environment.
> > > **High Scalability:** Given by agent count in Table 1.
> > > **Partial Observability:** Agents have a limited observation window compared to the size of the maps, forcing them to search for unknown information.
> > > **Flexible Observations and Actions:** Given the variety of observation types: images, minimaps, text encodings, and raw ground truth data, and the variety of action types: low-level actions, primitives, and high-level task descriptions via the Execution Module.
> > > **Heterogeneous Agents:** Given by the 4 different agent types.
> > >
> > > ## Point 2: Mechanisms of Communication
> > >
> > > *“If the communication is algorithm-side, do you mind responding to my point about semantic consistency with wildfire responses?”*
> > >
> > > *“It does beg the question then, how the different entities are meant to communicate in a manner that is semantically consistent with wildfire responses. For example, are the agents allowed to communicate with each other on the algorithm-side (as opposed to communicating through the environment) as per the pseudocode? Is this meant to simulate radio communication?”*
> > >
> > >
> > > Since the communication is handled by the algorithm-side, semantically consistent communication with wildfire response must be hard-coded to ensure such consistency. By default, all agents can communicate with other agents, insofar as the current baseline allows, with no delays/loss/or distance restrictions. Observing the robustness of multi-LLM algorithms with environmental communication restraints is definitely an interesting idea we would like to cover in future work. To our best knowledge, no prior research has comprehensively tackled this problem under similar settings as ours due to the scale and abstraction limits of prior environments. We will note this in our Future Work section in the revised paper.

---

> ### Author Response · Authors · 2025-10-25
> **Response to Reviewer qfL5 cont.**
>
> ## Point 3: Behavioral Goals
>
> *“The paper lists what the behavioral goals are, but in the paper (including table 8), there is no enumeration of which behavior goals correspond to which tasks/levels.”*
>
> Table 2 shows which behavioral goals are assigned to each task/level in the behaviors column. The acronyms are defined in Behavioral Goals.
>
> *“However, when computing the BCS, there is no clear way to decouple the degree to which each task requires competency in each behavioral goal. This can lead to a large amount of correlation that makes it extremely difficult to know, or even believe, whether the BCS is capturing the behavior meant to be measured.”*
>
> While it is true that multiple levels may share and be associated with multiple behavioral goals, it is inevitable with high-level abstract concepts. We cannot make the assumption that the behavioral goals are mutually exclusive, nor can we design levels that strictly test only one behavior without trivializing the complexity of our tasks. However, we do ensure that no two behavioral goals are tested on the exact same set of levels, preventing strict correlation. Behavioral goals allow us to use the performance scores on our benchmark to quantitatively compare and contrast frameworks across diverse general behavioral skills.
>
> *“As a side note, it is not clear the maximum score of 0 is indeed accurate for open-ended tasks. Is it possible to achieve a score of 0 on the open-ended tasks with negative penalties?”*
>
> For open-ended tasks, 0 is given as a hypothetical maximum score because there are only negative penalties. However, for these open-ended tasks, we logarithmically scale the scores to amplify the smaller improvements over the baseline score, versus the percentage towards this hypothetical maximum.
>
> ## Point 4: Seeds and Hyperparameters
>
> *“Only 3 seeds are run for everything, which definitely hurts the convincingness of the results.”*
>
> As shown in Figure 7, token costs scale poorly as agent count increases. Since our experiments mostly involve a large number of agents, cost is not a trivial problem to running more seeds. However, given how all baselines fail consistently to make even preliminary progress in larger and complex tasks, we believe that it’s reasonable to conclude that our claims on current performance and remaining challenges still hold.
>
> *“A.18 Random Seeds and Other Hyperparameters" (no hyperparameters are mentioned?)”*
>
> *“The pseudocode seems to suggest that the baselines do not have hyperparameters. Clarifying whether this is the case is helpful for understanding.”*
>
> In A.18, we wrote that all LLM temperatures were set to 0, and with one completion per decision step. However, we agree that we should have included the LLM model and the token context window. As for baseline parameters, indeed, most baselines do not have that many relevant hyperparameters, since there is no learning. However, some baselines do have some that we did not mention, such as the number of communication rounds per timestep and the lifetime of messages in some baselines. We will include this in the revised paper. Thank you
>
> ```
> EMBODIED Communication Rounds per Timestep: 2
> COELA Max Messages: 30 Messages
> EMBODIED Message Lifespan: 3
> (COELA uses max messages and EMBODIED uses lifespan because COELA has a single continuous chat while EMBODIED has independent chats that should be synced)
> ```

---

> ### Author Response · Authors · 2025-10-25
> **Response to Reviewer qfL5 cont.**
>
> ## Miscellaneous Questions
>
> *It appears like there are 16 levels in tables 2 and 9. Clarifying whether there are 12 or 16 levels is important, as the entire paper's empirical results hinge on this.*
>
> There are 12 distinct levels; however, 4 of them have multiple sizes. We do not count them as additional levels because they test the same task with the same scoring function and the same behavioral goals.
>
>
> *“The costs of running an environment are naturally of critical importance for the community and readers to understand whether the environment should be used. Figure 7 does not make it clear how many API calls come from the baseline methods versus actually making the API calls for running the environment itself. Clarification here is key.”*
>
> We agree that the costs of running an environment cannot be overlooked. Figure 7 only shows the API calls from the baselines because there are no API calls for running the environment, which is entirely self-contained within our contributed platform CREW-Wildfire. We will open-source the entire stack for the community to build upon, so there will be no API cost on the environment. We did, however, measure the computational costs and resource usage of running the CREW-Wildfire environment up to the full 1,000,000 cells and 2000 agents. The results are located in A.15.
>
> *“I am looking at the pseudocode for HMAS, Embodied Implementation, COELA, and CAMON. It does not appear that they have any learning. If there is no learning, I think it is important clarify in the text.”*
>
> Yes there is no learning in the baselines, which is consistent with their original implementations. These baselines are fully zero-shot, LLM-based, multi-agent Agentic AI frameworks with no inference-time fine-tuning. We will clarify this in the revised paper.
>
>
> *“Should use \citep (instead of cite or citet)”*
>
> *“A.17: ‘compentency’ should be ‘competency’”*
>
> *“In the execution model prompt on page 20, ‘PRIMATIVES" should be "primitives’”*
>
> *“Fig. 7 is referenced, seemingly out of place, in section 4. Resolving this would benefit the clarity”*
>
> Thank you for bringing these to our attention. We will fix these in the revised paper.

---

> > ### Comment · Reviewer_qfL5 · 2025-11-03
> > **Response to Miscellaneous Questions**
> >
> > Thank you! This response clarifies quite a bit. I do have a couple questions, though.
> >
> > > there are no API calls for running the environment, which is entirely self-contained within our contributed platform CREW-Wildfire.
> >
> > Q: The environment corresponds to everything in (a) in Figure 2 correct?
> >
> > Q: All the algorithms use the same perception and execution models, correct?

---

> ### Comment · Reviewer_qfL5 · 2025-11-03
> **Response to Points 3 and 4**
>
> Thank you again for your response.
>
> ## Point 3
>
> > While it is true that multiple levels may share and be associated with multiple behavioral goals, it is inevitable with high-level abstract concepts. We cannot make the assumption that the behavioral goals are mutually exclusive, nor can we design levels that strictly test only one behavior without trivializing the complexity of our tasks. However, we do ensure that no two behavioral goals are tested on the exact same set of levels, preventing strict correlation. Behavioral goals allow us to use the performance scores on our benchmark to quantitatively compare and contrast frameworks across diverse general behavioral skills.
>
> I empathize here, and understand that perfection cannot be achieved when seeking complex environments that test a multi-faceted set of skills.
>
> However, as much as I would like it not to, my concern still stands.
> 1. All tasks are weighted equally. This has two issues. The first is that the task difficulties are not necessarily the same, and achieving good performance at one skill may be intimately tied to performance in other skill. This is the correlation that has already been pointed out. The second point, related to the first, is that this method of measurement implicitly indicates that each task requires similar amounts of a certain behavior. These two issues certainly pollute the semantics of the metric.
> 2. The normalization scheme does not seem to account for task difficulty (I don't know how this can be done).
> 3. The BCS has no validation as a metric in and of itself. I am not saying every metric needs validation, but when combined with the above two points, I struggle to have confidence in what it measures. It may serve as useful for comparing algorithms perhaps, but we do not know that.
>
> Thank you for clarifying the other points. They indeed help my understanding of the paper and are sensible.
>
> ## Point 4
> > As shown in Figure 7, token costs scale poorly as agent count increases. Since our experiments mostly involve a large number of agents, cost is not a trivial problem to running more seeds.
>
> This is understandable, but this argument can then be used by any other paper attempting to publish using this environment. If it is prohibitively expensive to run more than a few seeds, that seems to be a severe detractor of this being of interest to the community. We wouldn't want every paper utilizing this environment to simply run 3 seeds (though hopefully they would have better scaling properties)
>
> Could more seeds not be run at least with the lower agent counts?
>
> Do you mind sharing what the costs are like to run a standard algorithm for this environment?
>
> > However, given how all baselines fail consistently to make even preliminary progress in larger and complex tasks, we believe that it’s reasonable to conclude that our claims on current performance and remaining challenges still hold.
>
> This is a fair point.
>
> > However, some baselines do have some that we did not mention, such as the number of communication rounds per timestep and the lifetime of messages in some baselines. We will include this in the revised paper. Thank you
>
> Excellent, thanks.
>
> > ...since there is no learning...
>
> Do most methods in this space not use learning? I am genuinely curious to try and better understand.

---

> > ### Author Response · Authors · 2025-11-06
> > **Response to Points 3**
> >
> > ## Point 3: Behavioral Goals
> >
> > *“All tasks are weighted equally. This has two issues. The first is that the task difficulties are not necessarily the same, and achieving good performance at one skill may be intimately tied to performance in other skill. This is the correlation that has already been pointed out. The second point, related to the first, is that this method of measurement implicitly indicates that each task requires similar amounts of a certain behavior. These two issues certainly pollute the semantics of the metric.”*
> >
> > *“The BCS has no validation as a metric in and of itself. I am not saying every metric needs validation, but when combined with the above two points, I struggle to have confidence in what it measures. It may serve as useful for comparing algorithms perhaps, but we do not know that.”*
> >
> > We completely understand your concerns. Since each task for each behavioral goal calculation is weighed equally, it is difficult to say where the value of BCS comes from. Moreover, since the “difficulty” differs across levels (although not quantitatively), a BCS for a behavior may be simply low due to the difficulty of the levels it is associated with. We would, however, like to note that the observation ”each task requires similar amounts of a certain behavior” is not accurate since the behaviors are calculated by the tasks, not the other way around. Regardless, the previous concern still stands.
> >
> > Given these concerns, we would like to explicitly list what can and what cannot be inferred from BCS scores to help clarify this. We will also include this discussion in our revised paper..
> >
> > What can be inferred:
> > - The relative comparison between algorithms on the same behaviors, ie. algorithm $A_1$ performing better at tasks with behavior $B_1$ than algorithm $A_2$. Since they are evaluated on the same levels, this is a valid claim.
> > - The relative comparison between BCS scores of the same algorithm, ie. algorithm $A_1$ performing worse at tasks with behavior $B_1$ versus $B_2$. While this may be a result of tasks with behavior $B_1$ being more “difficult”, we are not claiming that this is not the case. Instead, this should hint at the behavior perhaps being the reason why it is more difficult.
> >
> > What cannot be inferred:
> > - The meaning of any individual BCS score of a given algorithm on a given behavior on its own. As you noted, due to the range of levels it may be evaluated on, with no normalization of difficulty, the BCS score may not be a precise representation of that algorithm's proficiency in exclusively that behavior. Developing a more fine-grained evaluation system remains a open question and we will note this as future work.

---

> > > ### Author Response · Authors · 2025-11-06
> > > **Response to Point 4**
> > >
> > > ## Point 4: Seeds and Hyperparameters
> > >
> > > *“This is understandable, but this argument can then be used by any other paper attempting to publish using this environment. If it is prohibitively expensive to run more than a few seeds, that seems to be a severe detractor of this being of interest to the community. We wouldn't want every paper utilizing this environment to simply run 3 seeds (though hopefully they would have better scaling properties)”*
> > >
> > > *“Could more seeds not be run at least with the lower agent counts?”*
> > >
> > > This is a good point. We agree and also hope that, perhaps by this benchmark, the community will be inspired to build algorithms that intentionally scale efficiently, since up until now, there were no benchmarks that really required them to due to the limited scale. The baselines tested in this benchmark were clearly not made for the scale of our whole environment. However, we believe that it is possible to enhance the efficiency of the algorithms. We won’t attribute the inefficiency of the existing algorithms with respect to the cost to our benchmark design, since the inefficiency is not led by our benchmark formulation or design. Rather, our work follows the state-of-the-art implementations and reports their performance and efficiency. Increasing algorithm efficiency is out of our scope.
> > >
> > > We are also happy to share that we ran additional experiments on the smaller levels. We ran the following levels for each baseline for a total of 20*4=80 tests:
> > > 5 Cut Tree (small), 3 Cut Tree Lines (small), 5 Scout Fire (small), 2 Scout Fire (large), 3 Rescue Civilians Known (small), 2 Rescue Civilians (large)
> > >
> > > Here are the results with standard deviation per level.
> > >
> > > | Environment                              | CAMON        | COELA        | Embodied     | HMAS_2       |
> > > |------------------------------------------|---------------|---------------|---------------|---------------|
> > > | Cut_Trees_Lines_small                    | 30.00±0.00    | 30.00±0.00    | 30.00±0.00    | 28.00±2.65    |
> > > | Cut_Trees_Sparse_small                   | 18.00±0.00    | 14.00±4.18    | 14.60±2.07    | 17.40±0.89    |
> > > | Rescue_Civilians_Known_Location_large    | 7.00±0.00     | 4.00±1.41     | 5.00±0.00     | 5.00±0.00     |
> > > | Rescue_Civilians_Known_Location_small    | 3.00±0.00     | 0.67±0.58     | 2.67±0.58     | 2.33±0.58     |
> > > | Scout_Fire_large                         | 0.00±0.00     | 0.00±0.00     | 0.00±0.00     | 0.00±0.00     |
> > > | Scout_Fire_small                         | 1.60±0.89     | 0.00±0.00     | 0.80±0.84     | 1.20±1.10     |
> > >
> > >
> > > This data accurately matches our previous findings, and we will aggregate this data in our final results in Table 3. We will also recalculate our BCS data with this updated data. Here is the full data and costs: https://postimg.cc/G4gpY1xp
> > >
> > > *“Do you mind sharing what the costs are like to run a standard algorithm for this environment?”*
> > >
> > > The costs per experiment range from \$0.5 to \$20, depending on the baseline, agent count, and level. See the table linked above for the costs per experiment on the low-agent count levels. Since per experiment per seed on more complex levels can cost up to ~\$20, we did not run more seeds for them and stayed with 3 seeds per experiment. As noted earlier in our response, we believe that this cost is not attributed to our benchmark as the key contribution of our work. The limitation is the low efficiency of existing algorithms. We hope that our work will inspire more studies to consider efficiency when the problem complexity and scale increases.
> > >
> > > *“Do most methods in this space not use learning? I am genuinely curious to try and better understand”*
> > >
> > > Within the multi-agent space, many Multi-Agent Reinforcement Learning (MARL) or Imitation Learning approaches leverage learning, however, among the recent development of LLM/Foundation-Model based approaches for multi-agent Agentic AI framework (our focus of this paper), learning is typically not used.

---

> > > ### Comment · Reviewer_qfL5 · 2025-11-10
> > > **Response**
> > >
> > > Thanks for your responses. I'm still going through everything, but I should say here:
> > >
> > > > We would, however, like to note that the observation ”each task requires similar amounts of a certain behavior” is not accurate since the behaviors are calculated by the tasks, not the other way around.
> > >
> > > Indeed, I misspoke, but what I meant was each task that has a certain behavior goal. Each task with the same behavior goal is given equal weighting.

---

> > > > ### Author Response · Authors · 2025-11-12
> > > > **Response**
> > > >
> > > > *"Indeed, I misspoke, but what I meant was each task that has a certain behavior goal. Each task with the same behavior goal is given equal weighting."*
> > > >
> > > > Yes, you are correct in that each eligible task is given equal weight in the calculation of a BCS score for a given behavior. However, as with our previous response, we claim that our inferences still stand: the relative comparison between algorithms on the same behaviors and the relative comparison between BCS scores of the same algorithm. Since the tasks are of equal weight, the BCS score gives us the average score a given algorithm has on tasks with such behavior.

---

### Review · Reviewer_vdyu · 2025-10-29

**Summary Of Contributions:**

Authors introduce an open-source benchmark built atop the CREW simulation platform, designed to evaluate scalability, robustness, and coordination capabilities in complex, dynamic environments allowing for the procedural generation of response scenarios (Use case chosen is wildfire fighting).
Among its features  are terrain map generation and support for different types of agents (4 introduced).
The paper claim is to have built a benchmark specifically designed to evaluate Agentic multi-agent LLM systems under conditions of real-world scale and complexity. The main benchmark features are the support for:
      - Heterogeneous agents (4 types introduced, 2000+ handled),
      - Complex and dynamic terrain (1 million+ cell maps),
      - LLM-compatible multi-agent framework,
      - Perception and Execution modules that enable language-based agents to interpret sensory information and generate executable actions,
      - Realistic objectives under:
              o Partial observability,
              o Stochastic environments.

**Audience:**

Yes

**Audience Explanation:**

This wok constitutes a good instantiation effort of the CREW platform introduced in a separate paper previously published in this venue.

**Broader Impact Concerns:**

No.

**Claims And Evidence:**

Yes

**Claims Explanation:**

In the evaluation use case (Wildfire fighting simulation) fire dynamics are represented through cellular terrain decomposition enhanced with slope, wind, vegetation type, moisture etc.
The challenge in such evaluations is to increase number and diversity of agents, complexity of environments that they interact in, with complex perception capabilities that could be transformed int textual communication elements.
An interesting start could be in repurposing of the current use case to fundamentally similar situations (health care management of contagious diseases for example) allowing to minimize context and environment setup time and cost and the reuse of trained models.

The limitations section is well laid and enumerates the various simplifications needed at this stage to evaluate the proposed benchmark but also traces paths to future improvements.

**Requested Changes:**

Nevertheless, a few aspects need clarification:
- In “3.2 Environment Design: Wildfire Simulation”:
      o Q1: How does θslope scale this influence based on terrain steepness?
      o Q2: While the wind vector w⃗ represents the wind speed and direction, what does the  ∆⃗x  direction vector magnitude represent
      o “Moisture ratio attenuates the likelihood of ignition in wetter regions”:
              Q3: Does this make it binary?
              Q4: Any baring on propagation speed?
- In “3.3 Agent Design”:
      o “Observations support third-person agent image data”:
              Q5: Care to clarify?
- In “3.3.1 Perception and Execution Modules”
      o “Given that our core environment requires precise, discrete and continuous control, we developed”:
              Q6: Can you please detail how these 3 aspects (precision, discretization and continuous control) are required or present them selves in this benchmark?
- In “3.4 Pillars of CREW-Wildfire Design”:
      o “Task complexity” seems a misnomer:
              Q7:  Shouldn’t the benchmarking suite pillar be adaptation to task complexity in the simulation scenario?
- In “4 CREW-Wildfire Benchmarking Suite”:
      o “Where an algorithm is conceptually challenged”:
              Q8:  Care to clarify?
- In “Procedurally Generated Task Levels”:
      o “We define 12 procedurally generated levels (Fig. 7)”:
              Q9: Is this the right reference or Table 2?
              Q10: Are these 12 or 16 levels? (Same for Table 3)
- In “3.4 Pillars of CREW-Wildfire Design”:
      o “For heterogeneous agents, role heterogeneity demands coordination through complementary capabilities”:
              Q11: As the various defined agents have a set of shared capabilities (‘Move forward etc.), How are those leveraged?
- In “A.2 Perception Module”:
      o “AGENT X, and your current location is POSITION, and thus your minimap view will be the range X : [x0 − x1], Y : [y0 − y1]”:
              Q12: Aren’t minimap end coordinates [x0 – xi], Y : [y0 – yi], i instead of 1 where i depends on Agent’s minimap defined observation range?
- In “5.3 Behavioral Analysis”:
      o “Step 1: Level Normalization. The range of raw scores across different task levels is different due to different scoring functions. “:
              Q13: Normalization is generally applied for comparison reasons of variables with different ranges/scales. When using different scoring functions, does this “normalization” (or mapping) sill yield interpretable results ?
- In “3.4 Pillars of CREW-Wildfire Design”:
      o “tackling the inherent challenges that a large and complex cooperative task, such as wildfires creates”.
              Clarification: Isn’t the task rather “wildfire fighting”?

---

> ### Author Response · Authors · 2025-11-01
> **Response to Reviewer vdyu**
>
> We thank the reviewer for the thoughtful and positive comments and your recognition of our contributions. We would like to address all of your questions with point responses:
>
> ## In “3.2 Environment Design: Wildfire Simulation”:
> *“Q1: How does θslope scale this influence based on terrain steepness?”*
>
> Fire spreads more easily going upwards than downwards, so the terrain's steepness plays a role in the spread. However, looking deeper at our equation in 3.2, we realize we made a typo, which may have caused confusion. Our current equation implies that fires cannot spread downward due to the multiplication. Instead, the $\\theta_{slope}$ should have been replaced as a “slope factor” which is given by this equation:
>
>
>
> $slopefactor(\\theta) =
> \\begin{cases}
> \\dfrac{e^{-k \\theta}}{2e^{-k \\theta} - 1} & \\text{if } \\theta < 0 \\\\
> e^{k \\theta} & \\text{if } \\theta \\ge 0
> \\end{cases}$
>
>
> This is what the slope factor looks like across different $\\theta$ and a fixed $k$:
>
> https://postimg.cc/2bQ8sFzB
>
>
> Thank you for pointing out the confusion with this equation. We will fix this typo in the revised paper.
>
> *“Q2: While the wind vector w represents the wind speed and direction, what does the ∆x direction vector magnitude represent?”*
>
> The ∆x direction vector represents the vector from the source cell to the neighboring cell. The magnitude would be the distance between the two cells. 1 if they are adjacent, /sqrt{2} if they are diagonal.
>
> *“‘Moisture ratio attenuates the likelihood of ignition in wetter regions’:  Q3: Does this make it binary?”*
>
> The moisture ratio is not binary, but whether neighboring cells are ignited is binary.
>
> *“Q4: Any baring on propagation speed?”*
>
> The propagation speed is given by the game update frequency, which the user can set.
>
> ## In “3.3 Agent Design”: “Observations support third-person agent image data”:
>
> *“Q5: Care to clarify?”*
>
> Absolutely. Each agent has a virtual camera that streams third-person observations. Here is an example image showing some minimaps and third-person images (side view):
>
>
>
> ## In “3.3.1 Perception and Execution Modules”: “Given that our core environment requires precise, discrete, and continuous control, we developed”:
>
> *“Q6: Can you please detail how these 3 aspects (precision, discretization, and continuous control) are required or present themselves in this benchmark?”*
>
> To perform actions within the environment itself, agents need to use precise actions, such as moving to exact locations or spraying water in exact directions, to receive perfect scores since the scores in these tasks are based on the exact target locations. Moreover, the action space of the environment is also made up of both discrete and continuous parameters. Please consult A.3 to see the full list of action primitives. This complexity highlights the need for our Execution Module, which converts text commands to action primitives, since LLMs on their own struggle with precise control.
>
> ## In “3.4 Pillars of CREW-Wildfire Design”: “Task complexity” seems a misnomer:
>
> *“Q7: Shouldn’t the benchmarking suite pillar be adaptation to task complexity in the simulation scenario?”*
>
> We would like to clarify that, in this section, we discuss the CREW-Wildfire environment, rather than the specific skills required to address it. We agree that adaptation to task complexity would be an important skill needed to solve complex tasks due to the evolving nature of the tasks with uncertainty. However, the context in this discussion centers around the environment and has not yet reached solutions.
>
> ## In “4 CREW-Wildfire Benchmarking Suite”: “Where an algorithm is conceptually challenged”:
>
> *“Q8: Care to clarify?”*
>
> This refers to the Behavioral Goals Analysis presented in Section 5.3, which provides insights into where, on a behavioral level, algorithms succeed or fail. We will clarify this in our revised paper.
> In “Procedurally Generated Task Levels”: “We define 12 procedurally generated levels (Fig. 7)”:
>
> *“Q9: Is this the right reference or Table 2?”*
>
> Thank you for pointing out this typo. Yes, it is supposed to be Table 2. We will fix it in our revised paper.
>
> *“Q10: Are these 12 or 16 levels? (Same for Table 3)”*
>
> There are 12 distinct levels; however, 4 of them have multiple sizes. We did not count them as additional levels because they test the same task using the same scoring function and behavioral goals.

---

> > ### Author Response · Authors · 2025-11-01
> > **Response to Reviewer vdyu cont.**
> >
> > ## In “3.4 Pillars of CREW-Wildfire Design”: “For heterogeneous agents, role heterogeneity demands coordination through complementary capabilities”:
> >
> > *“Q11: As the various defined agents have a set of shared capabilities (‘Move forward etc.), How are those leveraged?”*
> >
> > Some capabilities and statistics are not shared among agents. For example, since helicopters cannot rescue civilians independently and firefighters cannot move quickly over long distances, helicopters and firefighters must work together to rescue civilians on large maps. Another example is that drone agents cannot cut trees, while firefighters and bulldozers cannot see fires from far away; therefore, drones and ground teams must work together to scout and address fires. All four agent types heavily depend on each other for even slightly more complex tasks.
> >
> > ## In “A.2 Perception Module”: “AGENT X, and your current location is POSITION, and thus your minimap view will be the range X : [x0 − x1], Y : [y0 − y1]”:
> >
> > *“Q12: Aren’t minimap end coordinates [x0 – xi], Y : [y0 – yi], i instead of 1 where i depends on Agent’s minimap defined observation range?”*
> >
> > Since the minimaps are centered around the agent’s location, x0 and x1 are actually calculated by x0 = x-range/2, x1 = x+range/2. Y0 and y1 are also calculated accordingly.
> >
> > ## In “5.3 Behavioral Analysis”:  “Step 1: Level Normalization. The range of raw scores across different task levels is different due to different scoring functions:
> >
> > *“Q13: Normalization is generally applied for comparison reasons of variables with different ranges/scales. When using different scoring functions, does this “normalization” (or mapping) still yield interpretable results?”*
> >
> > Yes, since we normalize the scores by comparing them to the objective worst and best possible scores, you can generally interpret the normalized score by how close the raw score is to the best or worst possible score. However, even these normalized scores are not necessarily meant to be compared across different tasks; rather, they are normalized to provide equal weight to each level when calculating BCS.
> >
> > ## In “3.4 Pillars of CREW-Wildfire Design”:  “tackling the inherent challenges that a large and complex cooperative task, such as wildfires creates”.
> >
> > *“Q14: Clarification: Isn’t the task rather 'wildfire fighting'?”*
> >
> > Thank you for the suggestion. We agree that this improves clarity. We will update this term in our revised paper.

---

> > > ### Comment · Reviewer_vdyu · 2025-11-25
> > > **Author responses**
> > >
> > > Thanks for your responses to the questions asked.
> > > This clarifies your approach in building this environment and the many aspects it tackles.
> > > Will certainly better help evaluating it.

---

> > > > ### Author Response · Authors · 2025-11-25
> > > >
> > > > Thank you very much for your follow-up comment and for taking the time to review our work. We’re glad the clarifications were helpful, and we truly appreciate your thoughtful evaluation. Please let us know if there is anything else we can provide to further assist your assessment.

---

### Decision · Action_Editor_Tuhy · 2025-11-27

**Recommendation:** Accept with minor revision

**Audience:**

Yes

**Audience Explanation:**

See response above.

**Claims And Evidence:**

Yes

**Claims Explanation:**

This paper introduces a new benchmark environment for evaluating LLM-based multi-agent systems in large environments with heterogeneous agents, partial observability, and stochastic dynamics. It is believed that effective solution methods should be able to effectively manage communication, coordination, perform long-term planning, and more. The benchmark is explicitly designed to be compatible with LLMs, with observations and environment details being encoded as language.

Overall, it has been agreed upon that the proposed problem does introduce challenges that are not available in other benchmarks (e.g., how challenging it is, its large scale), and that a part of the community would be interested in it.

Many concerns were raised during the discussion period and all reviewers agreed that the responses provided to the authors helped in better assessing the paper. I urge the authors to incorporate that feedback into the final version of the paper (thus, justifying the ‘accept with minor changes’ outcome). In particular, some of the concerns that were raised included the perception module, communication, and the proposed behavior competency score (BCS).

Personally, I do not consider that every concern was addressed, but I think that the value of the platform stands by its own, even if there are still questions about things like BCS. Part of science is to self-correct, and time will tell if some of these concerns (e.g., BCS consistency and reliability, or the claims about abilities agents are thought to need) will be correct. Nevertheless, that does not take away the value of introducing the benchmark to push the frontier of evaluation.

Finally, I do find it very problematic that evaluation is performed with only three seeds, and I personally do not find the justification of a high experimentation cost to be fully satisfactory as the inability to properly perform an experiment should inform which questions one asks. That being said, I did not consider the comparisons between different agents as the key point of the paper, instead I saw them as anecdotal evidence for the usefulness of the proposed benchmark, and in this context, I do think there’s a contribution to be shared with the community here.

---

> ### Author Response · Authors · 2025-12-08
> **Response to Action Editor Tuhy**
>
> We thank the Action Editor for their time, consideration, and support of our paper. We have uploaded our final camera-ready version. We here also summarize the changes we have made based on all reviewers' comments, our rebuttals, and the action editor's comments:
>
> # **Revisions Summary**
>
> ## **1. New Experimental Results**
>
> * Added VLM vs. Perception Module ablation study in the appendix comparing ASCII encoding against GPT-4o vision on 100 observations, using the JudgeLLM evaluation protocol.
> * Extended experimental runs to **20 seeds** for six small-scale levels (data collected; will be integrated into **Table 3**).
> * Added explicit **cost information**: experiments range from **$0.5–$20** depending on baseline and level complexity.
> * Clarified that **Figure 7** reports only algorithm-side API costs; the environment itself has **zero** API cost.
>
> ## **2. Implementation Details**
>
> * Added complete model specifications: **GPT-4o (gpt-4o-2024-08-06)** with a **128K** context window.
> * Added baseline-specific hyperparameters:
>
>   * *Embodied*: 2 rounds/timestep, 3-timestep lifespan
>   * *COELA*: 30 max messages
> * Explicitly stated that **all baselines are zero-shot**, with no learning during inference.
>
> ## **3. Table Improvements**
>
> * Clarified: *"12 distinct levels with 4 having size variants = 16 total configurations"* in text and in the **Table 2** caption.
> * Added a **Max Score** column to **Table 3** describing the theoretical maximum for each level.
>
> ## **4. Technical Corrections**
>
> * Fixed fire-spread equation: replaced `θ_slope` with the complete piecewise slope-factor function **f(slope)** showing uphill vs. downhill dynamics.
> * Added environmental parameter details: moisture ratio is continuous; ignition is binary; propagation speed is user-configurable.
> * Added three concrete examples of **heterogeneous agent cooperation** requirements.
> * Corrected minimap coordinate formula to show explicit centering.
>
> ## **5. Typo Fixes**
>
> * Corrected reference error: *"Fig. 7" → "Table 2"*.
> * Fixed typos: *"PRIMATIVES" → "primitives"*, *"compentency" → "competency"*, *"wildfires creates" → "wildfire fighting creates"*.
> * Added forward reference for *"conceptually challenged"* to Section 5.3.
> * Verified citation formatting throughout.
>
> ## **6. Future Work Changes**
>
> * Rewrote future work into three areas:
>
>   1. **Scalable Architectures and Efficient Algorithms**
>   2. **Adaptive Planning and Reasoning**
>   3. **Evaluation and Human-AI Teaming**
> * Added commitments to open-source trajectory data and exploration of human baselines.
>
> ## **7. Appendix Additions**
>
> * Added **BCS interpretation guidance** explaining normalization purpose and limitations.
> * Added **coordinate system clarification** in the perception module prompt.
>
> ---
>
> # **Summary of Changes by Section**
>
> | Section                    | Changes                                                          |
> | -------------------------- | ---------------------------------------------------------------- |
> | **3.2 Environment Design** | Fire-spread equation fix; environmental parameter clarifications |
> | **3.3 Agent Design**       | Added heterogeneous cooperation examples                         |
> | **3.4 Pillars**            | Terminology correction ("wildfire fighting")                     |
> | **4 Benchmarking Suite**   | Level-count clarification; "conceptually challenged" reference   |
> | **4.1 Experiment Setup**   | Zero-shot learning statement                                     |
> | **5.3 Results**            | Figure 7 caption updated with cost clarification                 |
> | **5.3 Outlook**            | Rewrote  future work                           |
> | **Table 2**                | Enhanced caption                                                 |
> | **Table 3**                | Added Max Score column                                           |
> | **Appendix A.2**           | Coordinate system clarification                                  |
> | **Appendix A.4**           | Typo fix                                                         |
> | **New: Before A.15**       | VLM ablation study                                               |
> | **Appendix A.16**          | BCS interpretation guidance                                      |
> | **Appendix A.17**          | Title typo fix                                                   |
> | **Appendix A.18**          | Complete hyperparameter specifications                           |

---

> > ### Author Response · Authors · 2025-12-11
> > **Resubmitted Final Version**
> >
> > We realized that the previous version was still anonymized, so we updated the template and resubmitted. Apologies for the confusion.